# Modelling of Environmental Ageing of Polymers and Polymer Composites—Modular and Multiscale Methods

**DOI:** 10.3390/polym14010216

**Published:** 2022-01-05

**Authors:** Andrey E. Krauklis, Christian W. Karl, Iuri B. C. M. Rocha, Juris Burlakovs, Ruta Ozola-Davidane, Abedin I. Gagani, Olesja Starkova

**Affiliations:** 1Institute for Mechanics of Materials, University of Latvia, Jelgavas Street 3, LV-1004 Riga, Latvia; olesja.starkova@lu.lv; 2SINTEF Industry, Forskningsveien 1, 0373 Oslo, Norway; Christian.Karl@sintef.no; 3Faculty of Civil Engineering and Geosciences, Delft University of Technology, P.O. Box 5048, 2600 GA Delft, The Netherlands; I.Rocha@tudelft.nl; 4Institute of Forestry and Rural Engineering, Estonian University of Life Sciences, 5 Kreutzwaldi St., 51014 Tartu, Estonia; juris.burlakovs@emu.ee; 5Faculty of Geography and Earth Sciences, University of Latvia, Raina Blvd 19, LV-1586 Riga, Latvia; ruta.ozola-davidane@lu.lv; 6Siemens Digital Industries Software, Via Werner von Siemens 1, 20128 Milan, Italy; abedin.gagani@siemens.com

**Keywords:** polymer, fibre, composite, biodegradable, environmental ageing, durability, modelling, mathematical modelling, multiscale modelling, diffusion, accelerated testing, lifetime prediction

## Abstract

Service lifetimes of polymers and polymer composites are impacted by environmental ageing. The validation of new composites and their environmental durability involves costly testing programs, thus calling for more affordable and safe alternatives, and modelling is seen as such an alternative. The state-of-the-art models are systematized in this work. The review offers a comprehensive overview of the modular and multiscale modelling approaches. These approaches provide means to predict the environmental ageing and degradation of polymers and polymer composites. Furthermore, the systematization of methods and models presented herein leads to a deeper and reliable understanding of the physical and chemical principles of environmental ageing. As a result, it provides better confidence in the modelling methods for predicting the environmental durability of polymeric materials and fibre-reinforced composites.

## 1. Introduction

Polymers and polymer composites are often exposed to elevated temperatures, mechanical stress, water and humid air environments, where their performance is negatively impacted by environmental ageing, reducing their service lifetime [1]. Fibre-reinforced polymer (FRP) composites are widely used in structural applications in the wind energy sector, marine and offshore, oil and gas, transportation and aircraft industries, where a design lifetime of composite structures is typically 25 years or more [2]. Hence, modelling and predicting the extent of composite material property deterioration due to exposure to water, humidity (e.g., open sea or rain), and other environmental stressors and media is of great interest for both the designers and end-users of composite materials and structures [3].

In the state-of-the-art, qualification of new composite materials is achieved mainly with experimental validation [2,4]. Such testing programs involve incredibly high costs, in which the verification of new composite materials is required (involving a massive testing programme) [2]. Even the slightest change in material or structural design choice involves extremely high testing costs which often is the “bottleneck” for the projects [2,4]. Certification costs can be as high as aircraft development costs in aviation programs, higher than USD 100 M [5]. Merely certifying a new composite material for aviation can cost up to several million USD as shown by a recent USD 3.7 M project led by the Korean Civil Aviation Certification Agency [6]. Furthermore, the timeline for material development ranges from a few years to modify existing materials to 10–20 years for new materials [7]. For the automotive industry, development cycles last an average of 20–30 months, and the cost of testing a new composite material would be higher than EUR 500,000 [8]. DNV GL has led a recent Joint Industrial Project to study offshore and marine applications composites, estimating potential savings for $10 M [9]. For wind blades, the impact of material certification can be evaluated in a similar magnitude; a recent cost model has found material and labour cost to comprise almost 75% of the total price of a composite blade [10].

Mathematical modelling has become an essential aspect of such validation [4]. Qualification by simulation, based on fundamental scientific principles, can substantially reduce the qualification costs. Thus, modelling tools are the future for the successful development of the polymer composite industry [2]. The current direction in the composite industry is to replace testing as much as possible with durability prediction models [2,11].

This review aims to systematize the state-of-the-art models, offering a comprehensive overview of the modelling tools for predicting environmental ageing and the degradation of polymers and polymer composites. Such systematization is seen as a step towards a fully functional modelling toolbox for predicting environmental durability, time-dependent properties and service lifetime of these materials. The review article is helpful for scientists and the industry alike for purposes of accelerated testing and is a step towards fewer testing efforts to reduce qualification costs of composite materials substantially.

Many approaches have been proposed to predict their long-term properties based on short-term measurements to reduce the time scales and costs required to obtain long-term data. The review would need to be lengthy and extensive to cover all types of models that have been developed. Therefore, this work is divided into two parts. This work (Part 1 of the review) covers Modular and Multiscale Methods. Furthermore, phenomenological modelling approaches for predicting long-term mechanical performance of polymer composites are covered in Part 2 of the Review [12].

This study is formatted as follows to achieve the objective of the review and to be of use to both academic and industrial readers. The modelling review starts with a broad contextual background provided in the Introduction. It is followed by the technical section on modelling. In the modelling section, a complex system of ageing models (microconstituents, modular and multiscale) is described. These include ageing and degradation models for three microconstituents, which are summarized in their respective subchapters: (1) Polymer/Composite Matrix, (2) Fibres/Composite Reinforcement, and (3) Sizing/Composite Interphase. When the microconstituents are in a composite, the degradation pathways of individual microconstituents are hard to distinguish [4,13]. This necessitates the introduction of modular and multiscale modelling approaches [2,4]. These approaches are covered in the “Modular and Multiscale Approach” section. The modelling section transitions into the section named “Emerging Trends” by offering an outlook on the current topics relevant in environmental ageing of polymer and composites. The discussed trends include biodegradation, biotic and abiotic degradation, degradation models in marine and compost environments, data-driven approaches of degradation, and the role of accelerated weathering and lab-based ageing versus field experiments. This section also explores the way a material-environment interaction can be a “two-edged sword” of an environmental phenomenon rarely discussed (Figure 1). In other words, in some cases, the environment negatively impacts material properties, whereas, in others, the material negatively impacts the environment. The determinant is the purpose of the material—there are applications where degradation is favourable (biodegradable plastics) and when it is unfavourable (structural polymers and composites). In the case of biodegradable polymers, both ends of life and valuable lifetime are shortened; thus, their application is only justified for specific applications (not for structural FRP). In conventional degradation tests (as mainly discussed within this paper), the aim is to approach or keep the material within a valid lifetime. In biodegradation tests (e.g., in compost or soil), the end of life (EoL) is simulated. Modelling approaches could be the same in both cases. However, the “safety” criteria would have to be replaced by the “decomposition” criteria.

The manuscript ends with section named “Environmental Economics Scope for Degradation Modelling”, showing how the technical aspects are strongly tied to the industrial and application scope of environmental modelling.

### Terminology

According to a standard terminology relating to plastics (ASTM D883) [14], degradation is defined as “a deleterious change in the chemical structure, physical properties, or appearance of a plastic”. Degradation affects mechanical, thermophysical, optical and other characteristics through plasticization, densification (embrittlement), cracking, colour changes, etc. Degradation is often considered to be a process of structural changes, of physical or chemical origin, taking place in polymers under the influence of an external factor that negatively impacts primary service properties [15]. Figure 2 shows the changes caused by the degradation process and the chemical and physicochemical factors that contribute to the degradation [16].

Polymer degradation can be classified as photo-oxidative, thermo-oxidative, ozone-induced, mechanochemical, hydrolytic, catalytic and biodegradation [16,17]. The term degradation could also be related to external factors resulting in properties deterioration, e.g., hydrothermal (elevated temperature and humid environment), environmental (combination of T, RH, UV), vibrational (dynamic load), etc. 

The degradation of polymers in marine environments is influenced by the complex interaction of various factors (temperature, moisture, pH, UV and active microorganisms) [18]. It has been proposed that the environmental impact of plastics entering the sea should be comprehensively assessed based on the following three main aspects: field assessment according to the environment in which the plastic waste is exposed (seawater composition such as salinity, water depth, the concentration of microorganisms and changes in water temperature, solar radiation depending on the season), assessment of biodegradability in a controlled laboratory environment to have the same conditions and chemical safety assessment [19]. ISO 22766 describes the method for evaluating the degree of decomposition in the sea. The technique involves exposing the plastic materials to actual field conditions in marine habitats [20]. ISO 22403, on the other hand, defines the assessment method for marine biodegradability. The plastic is degraded to carbon dioxide and water with the aid of marine microorganisms [21]. A detailed report recently provided a good overview of the definition of plastic biodegradation and testing, standards, and certification schemes [22].

A recent study highlights the investigation of various biodegradable polymers and an oil-based polymer as reference through laboratory testing and field trials in warm waters. The results assess the persistence of plastic objects in the event of their release into the sea [23]. In contrast to the relatively cold environment of the sea, the composting process takes place at higher temperatures (ca. 60 °C).

As far as industrially compostable plastics are concerned, the degradation process must strictly adhere to standard protocols. Although the existing standards are not perfect, they provide a solid basis for improving reproducibility in all test laboratories [24]. The ISO 14855 standard describes the method for biodegradation under controlled composting conditions [25]. It was developed to determine the biodegradability of plastic materials and is described in more detail in the literature [26,27]. Essential ISO and CEN standards related to biodegradability and compostability of plastics are described in detail in [28].

The extent of properties degradation depends on exposure time, the type of polymer or polymer composite, processing route (e.g., curing degree, annealing), the appearance of defects, pores, impurities, specific chemical groups, etc. Materials themselves could also be classified in terms of durability/reliability/lifetime requirements. FRPs are related to the class of high-performance composites used in automotive and aerospace industries, civil engineering, oil production, etc., with particularly high requirements related to durability and reliability to ensure structural integrity and safety during their long-term operation under environmental impact. Particulate composites and short-fibre reinforced polymers used for basic engineering applications are typically positioned as materials with moderate requirements on durability/reliability. In contrast, materials with a short-designed lifetime, e.g., commodity plastics, and especially food packaging plastics [29,30], need to be degraded relatively fast in the sea. Otherwise, there would be still a significant contribution to ghost fishing (large quantities of fish is found in lost gears) [31,32]. This aspect also includes fishing gear and net materials based on biodegradable instead of oil-based materials. Replacement of fossil-based to bio-based polymer composites is a global challenge nowadays focused on the development of novel polymer composites with tailored durability and biodegradability (see section Emerging Trends in Degradation Modelling of Biodegradable Polymers).

The valuable lifetime (service life) is the “actual period during which a structure or any of its components satisfy the design performance requirements without unforeseen major repair” [33]. It is a critical parameter for practical applications, serving as a manufacturer’s warranty for the material exploited in the service environment. After this time, the material reaches a threshold, i.e., acceptable property degradation, e.g., 50% of the initial value. The lifetime is closely related to the definitions of durability and reliability, i.e., the ability of an item to perform a required function under given environmental and operational conditions and for a stated period [34].

The term “ageing” is typically used when a material’s properties change in time. These changes could be related to the action of an external factor (e.g., elevated temperature, humidity—hygrothermal ageing) or could happen without any extrinsic influence due to inherent processes (structural relaxation—physical ageing, post-curing). Ageing does not always mean “degradation”, since in some cases, it could result in properties improvements, i.e., annealing [35]. Nevertheless, the inherent ageing processes are undesirable from the point of view of the material design, since these complicate and make modelling and prediction of material properties uncertain in the long-term, particularly under coupled action of several factors. Additionally, a link to biodegradable polymers must be made, for which “inherent” degradation is desirable and could be programmed. Nowadays, the prediction of the “end of life” of materials (not only “service life”) is an emerging trend that aims to reduce waste and increase the use of “climate-neutral” and sustainable materials.

Accelerated test methods (also called accelerated degradation tests) are programs that accelerate a material’s properties degradation by subjecting it to conditions outside its normal service range. In accelerated testing methodology, degradation is carried out in an intentional and controlled way aimed at practical reliability estimation in conjunction with modelling (statistical, constitutive, phenomenological) while reducing the time for experimental testing [36,37]. The service lifetime is predicted by modelling the evolution of the critical mechanical characteristics (e.g., strength, stiffness, creep compliance) under accelerated degradation and establishing the safety and reliability criteria.

## 2. Ageing and Degradation Modelling of Composite Microconstituents

There are three microconstituents in an FRP composite, namely, the polymeric matrix, fibres (reinforcement), and composite interphase. Composite interphase is formed on the surface of the fibres from a multi-component coating (sizing or size) and the matrix polymer during the manufacturing of a composite [11]. Each of the three microconstituents degrades differently and may also lead to coupling effects, affecting the degradation behaviour and rates of each other [4,11]. Therefore, when assessing degradation phenomena in composites, modular or multiscale approaches are preferred. The modular approach is described in this chapter and leads to the review of multiscale modelling methods in the following chapter. The individual modules are based on the individual microconstituents’ interactions with the environment and are summarized for each microconstituent in this chapter.

### 2.1. Polymer Degradation Models

#### 2.1.1. Predicting Polymer Properties from Chemical Structure

Quantitative Structure-Property Relationships (QSPR) relate the structural parameters of a molecule to its physical, chemical, and other properties [38]. There are various known QSPR-based and related approaches, such as Group interactions theory [39], Van Krevelen [40], Smooth Overlap of Atomic Positions (SOAP) approach [41], and Bicerano [38]. Alternatively, molecular dynamics (MD) techniques have been used but are usually more computationally heavy [42]. The physical and chemical properties of a polymer (or a composite matrix) can be calculated as regression functions of the connectivity indices [38]. The method is rooted in the idea of a topological graph theory, enabling a description of a polymeric system as several structural parameters—connectivity indices, which are derived from the chemical Lewis diagrams. The connectivity indices describe the electronic environment and the bonding configuration of each non-hydrogen atom in the respective polymer molecule. This approach is potentially helpful in building the link from the microstructure up to the macro properties [38]. The approach can be graphically summarized, as shown in Figure 3. 

The method is applicable to dry polymers, whereas extending it to plasticized conditions still poses a challenge, although some attempts that have been made, and it should eventually become possible. One of the possible links to the prediction from dry-to-wet is a Time-Temperature-Plasticization Superposition Principle (TTPSP), described in [43].

#### 2.1.2. Degradation Mechanisms

Environmental ageing may involve reversible and irreversible processes [44,45]. Irreversible processes are considered to be the ones that persist after redrying the polymer to its initial conditions. In contrast, reversible processes allow for the initial properties to be regained upon returning the polymer to its initial conditions, i.e., redrying the polymer to its initial water content [46]. Typical irreversible aging mechanisms of environmental ageing include the following [45,47,48,49,50]: (1) hydrolysis (chain scission); (2) thermo-oxidation (might involve chain scission, backbone modifications and/or thermo-oxidative crosslinking); (3) photo-oxidation (might involve chain scission, backbone modifications and/or photo-oxidative crosslinking); (4) residual curing (additional crosslinking); (5) leaching (initially present additives, impurities, or degradation products) [50,51].

Photo-oxidative degradation mechanism can be avoided in the absence of high-intensity light sources [52,53]. Thermo-oxidative degradation can be slowed down or avoided in the absence of heating sources [54]. Residual curing can be avoided if the polymer is appropriately and fully cured; it can be confirmed using experimental methods, i.e., Differential Scanning Calorimetry (DSC) [50,55].

Furthermore, the mechanical properties of various polymers are affected by diffusion, leading to a plasticization/swelling mechanism in the presence of water [2,4,48]. This negative effect is either reversible or irreversible, depending on the nature of the polymer [46].

#### 2.1.3. Diffusion and Leaching

Water uptake by polymers is governed by water diffusion and is influenced by the thickness of the specimen and the temperature of the environment [56,57,58,59]. When the polymer is in contact with humid or aqueous environments, water molecules can migrate into the polymer and negatively affect its properties [60,61]. Hydrophilic polymers are especially prone to this effect, whereas hydrophobic ones are more resistant [62]. Polymers, sorted by their hydrophobicity, are shown in Figure 4 [62].

Diffusion is an essential factor in the performance and environmental durability of plastics and FRPs, which undergo plasticization and swelling stresses [63]. Diffusion (and leaching, which is a reverse process) can usually be described for polymers using Fickian models [11]. For a majority of polymers and polymer composites, diffusion is the dominant mechanism of water uptake. It is represented by Fick’s 2nd law, which for orthotropic materials in Cartesian coordinates is:(1)$$\frac{\partial c}{\partial t}=\nabla \cdot \left(D\nabla c\right)=D_{11}\frac{\partial^{2}c}{\partial x^{2}}+D_{22}\frac{\partial^{2}c}{\partial y^{2}}+D_{33}\frac{\partial^{2}c}{\partial z^{2}}$$
where *c* (*x*, *y*, *z*, *t*) is the concentration of the diffusing water at a point with coordinates (*x*, *y*, *z*) in Cartesian space at time *t*, *D* is the positive definite symmetric matrix of diffusion coefficients *D*_ij_, and *D*_11_, *D*_22_ and *D*_33_ are the diffusion constants in directions 1, 2 and 3, respectively.

Fick’s 2 nd law for orthotropic materials in radial coordinates is
(2)$$\frac{\partial c}{\partial t}=D_{Z}\frac{\partial^{2}c}{\partial z^{2}}+D_{R}\frac{1}{r}\frac{\partial }{\partial r}\left(r\frac{\partial c}{\partial r}\right)$$
where *r* is the radial coordinate, *z* is the axial coordinate in radial space, $D_{R}$ and $D_{Z}$ are the radial and axial diffusion coefficients. 

Analytical solutions to Fick’s 2nd law for anisotropic materials can be found in work by Crank in [64]. Relations for calculating water contents *w*(*t*) are obtained by integration over the sample volume; these could be found elsewhere for orthotropic composite plates [57,65], rods [57,66], and pipes [57].

The directional diffusivities could be determined from weight gain curves for samples with sealed edges/faces by an impermeable material in order to eliminate water diffusion in two from three directions [67,68]. However, the reliability of this approach is limited due to the possible dissolution or degradation of a sealant during long-term exposure in water, especially under elevated temperatures. The more convenient way to identify the anisotropic diffusivity is based on the analysis of weight gain curves for samples of different sizes/fibre orientations that result in different contributions of water diffusion from certain sides of a sample [57,69,70]. These methods are often coupled with the concept of apparent diffusivity, taking into account anisotropy and edge effects [69,71,72]. For example, in the case of pultruded FRP rods, water absorption tests are conducted on samples of different lengths, while axial and radial diffusivities are determined by simultaneous fitting of sorption curves [57,66,73]. It is assumed that water uptake in long rods is dominated by radial diffusion, while additional water absorption due to axial diffusion through the ends is negligible. However, some uncertainties could arise from the definition of a “long enough” sample. A successive methodology for determining the directional diffusion coefficients in FRP rebars by using the apparent diffusivity is proposed in [66]. The diffusivities of an anisotropic material could also be estimated by applying structural micromechanical models if the diffusion parameters of the constituents (i.e., polymer matrix and fibres) are known [56,65,72]. The use of these approaches is justified for materials of “simple anisotropy”, such as unidirectional FRP. However, they could provide erroneous results due to simplistic assumptions on the composite structure and diffusion path. For complex composite structures, numerical/computational modelling micromechanical approaches are used [72]. More about these modelling approaches can be found in the chapter “Modular and Multiscale Approaches”.

In isotropic materials, the diffusion in all directions is characterised by only one coefficient *D*. Water content *w*(*t*) is obtained [57,64]:(3)$$\frac{w\left(t\right)-w_{0}}{w_{\infty }-w_{0}}=1-\frac{8}{\pi^{6}}\overset{\infty }{\underset{k=1}{\sum }}\overset{\infty }{\underset{n=1}{\sum }}\overset{\infty }{\underset{m=1}{\sum }}\frac{{\left[1-{\left(-1\right)}^{k}\right]}^{2}{\left[1-{\left(-1\right)}^{n}\right]}^{2}{\left[1-{\left(-1\right)}^{m}\right]}^{2}}{k^{2}n^{2}m^{2}}\exp\left(-\lambda_{k,n,m}^{2}Dt\right)$$
where $\lambda_{k,n,m}^{2}=\lambda_{k}^{2}+\lambda_{n}^{2}+\lambda_{m}^{2}={\left(\frac{\pi k}{h}\right)}^{2}+{\left(\frac{\pi n}{b}\right)}^{2}+{\left(\frac{\pi m}{l}\right)}^{2}$, $w_{0}$ and $w_{\infty }$ are the initial and equilibrium water contents, respectively. 

For a 1D diffusion problem, i.e., when sample dimensions in one of the directions are significantly smaller than in two others, e.g., *h* << *b*, *l*, Equation (3) with $w_{0}=0$ is simplified to the following equation: (4)$$w\left(t\right)=w_{\infty }\left[1-\frac{2}{\pi^{2}}\overset{\infty }{\underset{k=1}{\sum }}\frac{{\left[1-{\left(-1\right)}^{k}\right]}^{2}}{k^{2}}\exp\left[-{\left(\frac{\pi k}{h}\right)}^{2}Dt\right]\right]$$

Equation (4) could be written in other, more convenient for calculations, form [57,74]
(5)$$w\left(t\right)=w_{\infty }\left[1-\exp\left(-7.3{\left(\frac{Dt}{h^{2}}\right)}^{0.75}\right)\right]=w_{\infty }\left[1-G\left(t\right)\right]$$
(6)$$G\left(t\right)=\exp\left[-7.3{\left(\frac{Dt}{h^{2}}\right)}^{0.75}\right]$$

For the early stage of water absorption, Equation (4) is simplified to [74]
(7)$$w\left(t\right)=4w_{\infty }\sqrt{\frac{Dt}{\pi h^{2}}}$$

Then, according to Equation (7), data plotted in the axes *w* vs. $\sqrt{t}$ create a straight line with slope *D*. For samples of different thicknesses, data are analysed in the axes *w* vs. $\sqrt{t}/h$. At the late water absorption stage, w(t) asymptotically approaches equilibrium. These are the two main prerequisites for *Fickian* diffusion. If the first condition (linearity of *w* vs. $\sqrt{t}$ at the early stage of diffusion) is valid in most cases, then the second condition is often not achieved, and the equilibrium stage is either postponed or could not be reached. This behaviour is known as *non-Fickian* diffusion. Degradation [13,50,51,67], molecular relaxation [75,76] and specific interactions between the polar sites of the host polymer and a penetrant [77,78,79,80] are among the main factors contributing to the anomalous water absorption. 

Two-phase sorption models describe Non-Fickian diffusion. The Langmuir two-phase model considers a free diffusion phase and a bound phase of a penetrant that does not involve diffusion. The model employs Fick’s law for the free-phase water so that the driving force of diffusion is the concentration gradient. In the Langmuir model, two additional parameters are introduced, namely, *α*—the probability of transition of a water molecule from combined state to free phase and *β*—the probability of transition of a water molecule from a free to a combined phase. The Langmuir model for $\alpha ,\beta \ll \pi^{2}D/h^{2}$ and taking into account Equation (6) is given as follows [81]
(8)$$w\left(t\right)=w_{\infty }\left[1-\frac{\beta }{\alpha +\beta }\exp\left(-\alpha t\right)-\frac{\alpha }{\alpha +\beta }G\left(t\right)\right]$$

As follows from Equation (8), at α≫β, i.e., when the free phase is dominant, the Langmuir model transforms into Fick’s diffusion model. 

Other two-phase models are based on a conceptually different approach when a polymer itself is considered a two-phase system consisting of a general phase, where the most moisture sorption takes place, and a secondary phase characterised by lower hydrophilicity compared to the first phase. The origin of such “heterogeneity” could be related to the material structure [82] (inherent or altered due to water penetration) or the difference in the mobility of water molecules due to their clustering [80,83]. It is assumed that the sorption in each of the phases is driven by the diffusion mechanism yet is characterised by different sorption parameters $w_{\infty 1}$, $D_{1}$ and $w_{\infty 2}$, $D_{2}$, respectively. Then, according to Equation (5), the total water content is given as follows [82]:(9)$$w\left(t\right)=w_{\infty 1}\left[1-G\left(D_{1},t\right)\right]+w_{\infty 2}\left[1-G\left(D_{2},t\right)\right]$$

The equilibrium water content for such a polymer system is $w_{\infty }=w_{\infty 1}+w_{\infty 2.}$ Two-stage diffusion could also be explained by the viscoelastic nature of polymers, namely plasticization, ageing and post-curing effects accelerated by absorbed water and resulting in additional water uptake. The relaxation effects are taken into account in different ways. For example, by means of time-dependent diffusivity or changing boundary conditions [84,85]. Berens and Hopfenberg [86] found that the additive contribution of the diffusion and relaxation processes resulted in the relaxation-driven diffusion model of the following form [76,86]
(10)$$w\left(t\right)=w_{\infty D}\left[1-G\left(t\right)\right]+w_{\infty R}\left[1-\exp\left(-t/\tau_{R}\right)\right]$$
where $w_{\infty D}$ corresponds to the primary diffusion-driven saturation level, while $w_{\infty R}$ is the secondary equilibrium water content specified by network relaxation; $\tau_{R}$ is the first-order relaxation time related to the diffusion Deborah number [76].

The coupled action of water diffusion and structural relaxation in polymer networks was suggested by Bao et al. [87]. Relaxation effects accelerating water absorption are introduced into the model through the relaxation coefficient *k* [75,87]:(11)$$w\left(t\right)=w_{\infty }\left(1+k\sqrt{t}\right)\left(1-G\left(t\right)\right)$$

The constant *k* in Equation (11) can also be related to the polymer hydrolysis and leaching of the degradation products or uncured components [51,67]. In the latter case, water diffusion is accompanied by weight losses, and *k* has negative values. It is worth noting that processes described by Equation (11) do not involve achieving the equilibrium state. Alternatively, water absorption accompanied by weight decrease could be characterised by the two-phase additive model Equation (9), assuming that the leaching of polymer components is driven by a similar water diffusion mechanism [13,50,51]. For both diffusion and leaching, saturation is expected. 

Another important aspect when measuring diffusivity and saturation levels in FRPs is the complicated nature of the competing processes [13]. The non-Fickian models provide good fits to the experimental data in the long term; however, Fick’s model remains the general model for calculating the diffusivities and primary equilibrium water contents of polymers and composites. In composites, deviations from the Fickian-type water absorption are often related to degradation processes at the fibre-matrix interface or delaminations [13,88,89,90]. According to a standard ASTM D5229, testing stops when the mass increases with time stops [91], i.e., a plateau is reached. However, for FRPs, the water uptake increases after that due to the hydrolytic degradation of the sizing-rich fibre matrix interphase [13,92]. If the water uptake experiments stop, as suggested by ASTM, then the long-term behaviour is not captured. This can be explained by the degradation of the composite interphase [13]. In order to capture this behaviour, complex modelling via mass balance is required [13], which is described in the Interphase degradation chapter.

For composite plates, the water uptake (and also leaching) via a simplified analytical solution can be calculated using the following Fickian diffusion expression [56,91]:(12)$$M_{\infty }=\frac{M_{\infty }^{m}(V_{m}+V_{i})\rho_{m}+M_{\infty }^{v}V_{v}\rho_{water}}{V_{f}\rho_{f}+(V_{m}+V_{i})\rho_{m}}$$
where $\rho_{m}$ is the matrix density, $\rho_{f}$ is the fibre density, $\rho_{water}$ is the water density, $V_{f}$ is the fibre volume fraction, $V_{m}$ is the matrix volume fraction,$\;V_{i}$ is the interphase volume fraction, $\nu_{v}$ is the void volume fraction ($V_{f}+V_{m}+V_{i}+V_{v}=1$), $M_{\infty }^{m}$ is the polymer water content at saturation and $M_{\infty }^{v}$ is the void saturation water content (100 wt%). M(t) is the water content, $M_{\infty }$ is the water saturation content, t is time, h is the thickness, and D is the diffusivity in the thickness direction of the plate.

For FRPs, the diffusion model might also include such parameters as the void content and the presence of delaminations. The situation is complicated by the fact that the diffusion parameters are influenced in the course of ageing, and the ageing itself is also affected by the changing diffusion rate [88]. This complicated interconnectedness is explained in more detail in a 2019 study by Gagani [88].

#### 2.1.4. Swelling and Plasticization

Another physical effect that occurs when composites are exposed to water or humidity is hygroscopic swelling, which consists of an increase in the volume of the material, directly proportional to the concentration of diffused fluid [93]. The main effect of water on the mechanical property deterioration of some polymers is hygroscopic swelling [46]. Swelling is a specific response accompanying moisture diffusion in polymers and polymer-based composites [47]. Swelling can cause mechanical stresses on the material depending on the boundary conditions applied [46,94].

Various studies have explored the swelling of FRPs [94,95,96,97,98,99,100,101,102,103,104] and textile composites [105,106]. The nature of hygroscopic swelling in the polymeric matrix was investigated in works [1,60,76,80,94,107]. The influence of swelling on the fluid diffusion in polymers was described in [96,102,108,109,110]. The micromechanical models to predict transverse swelling were introduced in [97,105,106].

Hygroscopic swelling of polymers is typically isotropic and linear with water content. It can be predicted analytically as [93]:(13)$$\varepsilon_{h}=\beta w$$
where $\varepsilon_{h}$ is the hygroscopic strain, β is the coefficient of hygroscopic expansion (CHE), and w is the moisture concentration. Linear strain behaviour has been observed experimentally with increasing moisture concentration for composites and polymers [80,93,96].

Generally, the kinetics of swelling follows the weight gain changes appearing in a common Fickian-type (swelling strain vs. square root of time) curve, although this could be of a two-stage behaviour in the case of relaxation-driven diffusion [76].

Since swelling does not follow the ideal mixing law [47], it is necessary to perform swelling experiments for the polymer or to find the polymer CHE in the literature [107,111]. The swelling strain increases linearly with increasing water concentration for both composites and polymers [93,96]. For orthotropic laminates, three CHEs; $\beta_{x},\beta_{y},\beta_{z}$ are needed to predict swelling [112].

Krauklis et al. (2019) have shown that the hygroscopic swelling of FRPs can be analytically predicted from the swelling of the polymer using a model based on linear elasticity [112]. The model correlated well with experimental results and was validated by comparison with a Finite Element (FE) analysis.

Transverse swelling strains can be predicted as a serial connection of fibre and matrix. For many engineering reinforcements (carbon fibres, glass fibres, etc.), the swelling is null, $\varepsilon_{f}$ = 0. Therefore:(14)$$\varepsilon_{y}=V_{f}\varepsilon_{f}+\left(1-V_{f}\right)\varepsilon_{m}=\left(1-V_{f}\right)\varepsilon_{m}$$
(15)$$\beta_{y}=\frac{\varepsilon_{y}}{W_{c}}=\frac{\left(1-V_{f}\right)\varepsilon_{m}}{W_{c}}=\left(1-V_{f}\right)\frac{W}{W_{c}}\beta_{m}$$
where $\varepsilon_{y}$ is the composite transverse swelling strain, *V_f_* the fibre volume fraction, $\varepsilon_{f}$ is the fibre swelling strain, $\varepsilon_{m}$ is the matrix swelling strain, W is the moisture content in the matrix, $W_{c}$ the moisture content in the composite, $\beta_{y}$ is the transverse swelling coefficient.

Axial swelling can be predicted by employing a parallel connection model of fibre and matrix. The swelling of the matrix, in this case, is strongly constrained by the stiffness of the fibres [112].
(16)$$\varepsilon_{x}=\varepsilon_{f}=\frac{\sigma_{f}}{E_{f}}=\frac{\sigma_{m}}{E_{f}}\frac{1-V_{f}}{V_{f}}=\frac{E_{m}\varepsilon_{m}}{E_{f}}\frac{1-V_{f}}{V_{f}}$$
(17)$$\beta_{x}=\frac{\varepsilon_{x}}{W_{c}}=\frac{\sigma_{m}}{W_{c}E_{f}}\frac{1-V_{f}}{V_{f}}=\frac{E_{m}\varepsilon_{m}}{W_{c}E_{f}}\frac{1-V_{f}}{V_{f}}=\frac{E_{m}\beta_{m}}{E_{f}}\frac{W}{W_{c}}\frac{1-V_{f}}{V_{f}}$$
where $\sigma_{m}$ is the stress in the matrix, $E_{m}$ is the stiffness of the matrix, $\sigma_{f}$ is the stress transferred to the fibre, $\varepsilon_{x}$ is the composite axial strain, $\varepsilon_{f}$ is the fibre axial strain, $E_{f}$ is the fibre stiffness, $\beta_{x}$ is the axial swelling coefficient.

Another physical effect observed when composites are exposed to low molecular liquids, such as water, is plasticization, resulting in an increased flexibility of the polymer chains. As the chain flexibility increases—softening of the polymer on the introduction of plasticizer—the lower is the glass transition temperature (*Tg*) [113]. For polymers, a *Tg* drop is generally attributed to an action of plasticization or deterioration (i.e., chain scission) [44]. Such a *Tg* decrease in case of plasticization can be estimated using the Time-Temperature-Plasticization Superposition Principle (TTPSP), if the shift function is known [43]:(18)$$\log a_{dry-to-plast}=\frac{-E_{A}}{2.303R}\left(\frac{1}{Tg{\;}_{plast}}-\frac{1}{Tg{\;}_{dry}}\right)$$
where $Tg{\;}_{dry}$ and $Tg{\;}_{plast}$ are glass transition temperatures of dry and plasticized material, respectively; $E_{A}$ is the activation energy; $a_{dry-to-plast}$ is the shift function.

However, plasticization not only leads to softening in a purely mechanical sense but may also lead to changes in diffusive, dielectric, thermal and other mechanical properties. Furthermore, changes in all these properties are connected to alterations in molecular mobility and correlate with one another to some extent [114].

Typically, a plasticizing agent acts as a lubricant for reducing intermolecular friction and an agent breaking inter-molecular bonds via solvation in a polymer network [114]. The degradation of the tensile strength of polymers can often be attributed to the plasticization/swelling action and deterioration of the polymer [44,115]. The percentage reduction in the tensile strength of epoxies is related to the hydrophilicity of the resin blend, which Hoy’s solubility parameter may measure for hydrogen bonding [116]. Some authors also report a significant reduction in the fatigue life of polymers due to plasticization/swelling action [48]. 

#### 2.1.5. Hydrolysis

Some polymers undergo irreversible degradation due to hydrolysis, e.g., polyamides [117]. Hydrolysis kinetics of polymers were described in detail in works by Mazan et al. [117,118,119]. They developed a multiscale model for predicting the mechanical properties after hydrolytic degradation. The methodology described should apply to other matrix materials that undergo hydrolysis. 

Mazan et al. have reported that hydrolysis induced chain scission and chemicrystalization were the two main mechanisms of property change [119]. Furthermore, they were able to model the mechanical deterioration behaviour of PA11 [117,118,119] based on QSPR principles of Bicerano [38] and Jacques model [120].

The hydrolysis of biodegradable polymers was studied by Vieira et al. in [121,122]. They found that the time-dependent decrease in the tensile strength *σ* of biodegradable PLA-PCL (in the form of polymer fibres) was directly related to the reduction of molecular weight *M**_n_*. Modelling hydrolysis as a first-order kinetic mechanism, the hydrolytic damage $d_{h}$ is defined as follows [121]:(19)$$d_{h}=1-\frac{\sigma_{t}}{\sigma_{0}}=1-\frac{M_{nt}}{M_{n0}}=1-e^{-ut}$$
where *u* is the degradation rate. Subscripts *t* and 0 are related to the corresponding parameters at current time *t* and initial values. This approach could be extended to other biodegradable polymers [123].

### 2.2. Fibre Degradation Models

The two most common reinforcement material types are glass fibres (GFs) and carbon fibres (CFs), see Table 1. However, also basalt fibres (BFs) and aramid fibres (AFs) have an increasing use case in the composite industry [124]. More recently, Natural Fibres (NFs) have attracted growing interest in the research community [125,126].

#### 2.2.1. An Introduction to Glass Fibre Degradation

As of today, the most common type of fibre reinforcement materials are GFs [124,127,128]. GFs are hydrophilic and are susceptible to degradation when exposed to water environments [129]. The fact that GFs degrade in aqueous environments has been known for years, at the least, since the early 1970s [130,131,132]. However, not all GFs degrade at the same rates. The environmental stability of GFs strongly depends on the type of glass material. Various kinds of GFs exist, such as AR (“Alkali Resistant”), E (“Electric”), E-CR (“Electric/Corrosion Resistant”), C (“Chemical”), A (“Alkali”), R (“Reinforcement”), S/S-2-glass (“Strength”) listed in the order of increasing mechanical strength (see Table 1) [124]. Mechanical properties of various unaged fibres are summarized in Table 1 [124]. The degradation of GFs due to environmental attack in water can severely decrease their mechanical properties over time due to the development of corrosion-induced defects—flaws, such as surface cracks [133], subsequently leading to a strength and modulus deterioration of respective GFRPs [134].

Alongside the type of glass, the effect of sizing on the degradation rates of GFs cannot be neglected. It was shown by Schutte in 1994 [135] and later confirmed by Renaud and Greenwood in 2005 [136] that unprotected GFs exposed to water lose their strength relatively quickly, especially if they are mechanically loaded. In a more recent study by Krauklis and Echtermeyer in 2018, it was found that the degradation of sized GFs was slowed down by around six-fold when compared to desized R-glass fibres (R-GF) [137]. In another study by Krauklis and Echtermeyer et al. in 2019, a stress-corrosion of R-GFs was experimentally tested, quantitatively describing the acceleration of GF hydrolytic degradation by mechanical stress [138]. These effects (sizing protection and acceleration by mechanical stress) were analytically modelled and described in [137,138]. An analytical toolbox consisting of seven modelling modules was presented in a recent work (2021) on the Modular Paradigm for GFRPs [139]. The seven developed modules of the Reinforcement Modular Group within the Modular Paradigm are presented in Figure 5 (divided into 3 Dry Property Modules and 4 Degradation Modules). The last four modules are of particular interest in this review and are covered in the Modelling chapter.

#### 2.2.2. Chemical Reactions during Hydrolytic Degradation of Glass Fibres

During glass-water interactions, several chemical reactions may occur, shown in Chemical Reactions (R1)–(R11), and summarized in various sources, including works by multiple research groups from 1979 to 2021 [131,138,139,140,141,142,143,144]:(R1)$$\left(\equiv Si-ONa\right)+H_{2}O\to \left(\equiv Si-OH\right)+OH^{-}+Na^{+}$$
(R2)$$\left(\equiv Si-OK\right)+H_{2}O\to \left(\equiv Si-OH\right)+OH^{-}+K^{+}$$
(R3)$${\left(\equiv Si-O\right)}_{2}Ca+H_{2}O\to 2\left(\equiv Si-OH\right)+2OH^{-}+Ca^{2+}$$
(R4)$${\left(\equiv Si-O\right)}_{2}Mg+H_{2}O\to 2\left(\equiv Si-OH\right)+2OH^{-}+Mg^{2+}$$
(R5)$$\left(\equiv Si-O-Al=\right)+H_{2}O↔\left(\equiv Si-OH\right)+\left(=Al-OH\right)$$
(R6)$${\left(\equiv Si-O\right)}_{2}Fe+H_{2}O\to 2\left(\equiv Si-OH\right)+2OH^{-}+Fe^{2+}$$
(R7)$${\left(\equiv Si-O\right)}_{3}Fe+H_{2}O\to 3\left(\equiv Si-OH\right)+3OH^{-}+Fe^{3+}$$
(R8)$$\left(\equiv Si-O-Si\equiv \right)+OH^{-}↔\left(\equiv Si-OH\right)+\left(\equiv Si-O^{-}\right)$$
(R9)$$\left(\equiv Si-O^{-}\right)+H_{2}O↔\left(\equiv Si-OH\right)+OH^{-}$$
(R10)$$SiO_{2}+2H_{2}O↔H_{4}SiO_{4}$$
(R11)$$MeCl_{x}\overset{H_{2}O}{\to }\left(Me^{x+}\right)+xCl^{-}$$

A chemical reaction (R10) can also be written as a combination of subsequent reactions (R12) and (R13), meaning that initially $H_{2}SiO_{3}$ is formed, which dissociates weakly and further reacts with water to form silicic acid [11]:(R12)$$SiO_{2}+H_{2}O↔H_{2}SiO_{3}$$
(R13)$$H_{2}SiO_{3}+H_{2}O↔H_{4}SiO_{4}$$

As shown in the Chemical reactions (R1)–(R13), various competing reactions occur simultaneously. Initially, these reactions occur at independent rates (Phase I). Later one process becomes limiting and dominates the behaviour (Phase II). Respective Phase I and II are explained in more detail in the following subchapters.

#### 2.2.3. Molecular Mechanism and Kinetics of Degradation

The molecular mechanism of interactions needs to be understood to model the complex process of hydrolytic degradation of GFs. Herein, the term degradation denotes all processes which lead to or affect the mass loss of the glass material by interacting with water, according to [145]. 

Most studies that explain environmental degradation mechanisms of glass materials are based on surface reactions, chemical affinity and diffusion [129,146,147,148,149,150]. Geisler-Wierwille et al. described a process-driven approach that involves the congruent dissolution of glass, coupled to the precipitation and growth of an amorphous silica layer at an inwardly moving reaction interface [147]. In a more recent study by Ma et al., a combination of chemical affinity for controlling the distribution of Si atoms among different alteration phases and the diffusion barrier for the release rate of glass modifiers was implemented [149]. However, dissolution experiments in existing studies are mainly performed with bulk silicate glasses, and GFs or GFRPs are rarely studied [141]. However, some studies on the mechanism of GF degradation do exist. The state-of-the-art mechanistic understanding of GF degradation is based on the works by Grambow et al. (2001), Hunter et al. (2015) and Echtermeyer and Krauklis et al. (2018 and 2019) [138,140,141,142,145]. The complex nature of GF degradation involves several parallel processes, namely, gel layer formation, dissolution of glass matrix constituents, alkaline and alkaline earth ion exchange, and neoformation of solid reaction products. Some of these reactions occur in the glassy state, while others lead to the leaching of the reaction products into the surrounding aqueous environment [140,151]. The degradation of GFs proceeds in two phases, the first is an initial disorderly Phase I and a subsequent steady-state Phase II. In the short-term phase (Phase I [138,141]), hydrolytic degradation involves competing processes (ion exchange, gel formation and dissolution) [140,141,142]. In the long-term (Phase II), hydrolytic degradation is governed by the glass dissolution mechanism and follows zero-order reaction kinetics [140,141]. Such kinetics depend on the glass surface area in contact with water, proportional to the fibre radius. As the dissolution continues, the radius decreases, resulting in the mass loss deceleration [141]. The cumulative mass loss of all ions released is what manifests in the radius reduction [141]. 

There are also a few studies that concentrate on the kinetics of GFs dissolution. For instance, Mišíková et al. have studied the effect of temperature on the E-glass fibre (E-GF) leaching kinetics in distilled water [152]. Bashir et al. studied the kinetics of the dissolution of E-GFs at a high pH by immersing single fibres and measuring the diameter change [153]. They concluded that the rate-limiting step was either the diffusion of hydroxide ions through the solution or the E-GF etching itself [153]. Krauklis and Echtermeyer presented an analytical model termed the Dissolving Cylinder Zero-Order Kinetics (DCZOK) model. The model predicts GF dissolution kinetics during long-term hygrothermal ageing of GF bundles and GFRPs at various environmental conditions (pH, temperature and mechanical stress) [137,138,141]. The model can predict the mass loss, fibre radius reduction and flaw growth kinetics during dissolution [137,141]. The dissolution of GFs inside composites is slowed down compared to GF bundles and is addressed in the analytical model [137]. 

#### 2.2.4. Modelling of Mass Loss and Radius Reduction of Glass Fibres

A few solid-state models exist that apply to the GF dissolution, such as the Contracting Cylinder model [154], Shrinking Cylinder model [153] and Dissolving Cylinder Zero-Order Kinetic (DCZOK) model [141]. The latest model differentiates between the complex short-term (Phase I) and dissolution-dominated long-term (Phase II) stages. Furthermore, the DCZOK model describes mass loss, radius reduction and crack growth kinetics due to glass dissolution without the necessity of introducing additional terms such as a conversion factor; it relates the evolution of conversion to time. It was also shown to be linked to strength loss evolution [145]. Mass loss, fibre radius, crack growth and strength loss kinetics, all based on the DCZOK model (four degradation modules, see Figure 5), were systematized and discussed in a recent paper on the Modular Paradigm for composites [139].

The DCZOK model involves certain assumptions. For example, this model is deterministic, all fibres are assumed to have the same initial radius and the cross-sectional surface area at the end of the fibres is assumed to be negligible in calculations of the surface area. The length and density of the GFs are considered to be constant during the whole dissolution process. The model should be applicable to other types of glass, as SiO_2_ is the major component in virtually all types of glass [153], and SiO_2_ dominates the dissolution process, or at least does so in Phase II. The DCZOK model can be used to describe dissolution for each element separately and for the total mass loss [141]. The analytical DCZOK model equations for radius reduction during Phase I and Phase II are depicted in Equation (20) [141]:(20)$$\{\begin{array}{c} t\leq t_{st}:\;\;\;\;r\left(t\right)=r_{0}-\frac{K_{0}^{I}\xi_{sizing}}{\rho_{glass}}t\;\\ t>t_{st}:\;\;\;\;r\left(t\right)=r_{t_{st}}-\frac{K_{0}^{II}\xi_{sizing}}{\rho_{glass}}\left(t-t_{st}\right) \end{array}$$
where r is the GF radius after time t; $r_{0}$ is the initial GF radius; $\rho_{glass}$ is the density of GF; $K_{0}^{I}$ and $K_{0}^{II}$ are the rate constants for Phase I and Phase II, respectively; $\xi_{sizing}$ is the protective effect of sizing; $r_{t_{st}}$ is the GF radius after time $t_{st}$ when steady-state is reached.

GF dissolution degradation is an energy activated process and follows the Arrhenius principle–the rate of dissolution increases as the temperature increases [138]. However, the most dramatic environmental influence on the rate of GF dissolution is pH [138]. The dissolution rate is slowest at conditions close to neutral and accelerates towards both the acidic and basic ends, especially so in an extremely acidic environment. Environmental factors, such as acidity, temperature and mechanical stress all affect the material-environment energy-activated interactions, thus affecting the dissolution rate constants $K_{0}$. The DCZOK model accounts for the environmental conditions (*pH*, *T*, *σ*):(21)$$\frac{\partial m}{\partial t}=2n\pi l\left(r_{0}K_{0}\xi_{sizing}-\frac{{\left(K_{0}\xi_{sizing}\right)}^{2}}{\rho_{glass}}t\right)$$
(22)$$\frac{\partial m}{\partial t}=2n\pi l\left(r_{0}Ae^{-\frac{E_{A}\left(pH,\sigma \right)}{RT}}\xi_{sizing}-\frac{{\left(Ae^{-\frac{E_{A}\left(pH,\sigma \right)}{RT}}\xi_{sizing}\right)}^{2}}{\rho_{glass}}t\right)$$
where m is a total cumulative mass dissolved after time t; n is the number of GFs; l is the length of GFs; $K_{0}$ is the rate constant; A is the pre-exponential factor; R is the gas constant being 8.314 J/(mol∙K); T is the absolute temperature; $E_{A}$ is the activation energy; pH is the acidity; σ is the mechanical stress.

Complete analytical expression of the DCZOK model is:(23)$$\{\begin{array}{c} t\leq t_{st}:\;\;\;m_{dissolved}=n\pi l\left(2r_{0}K_{0}^{I}t-\frac{K_{0}^{I\;2}}{\rho_{glass}}t^{2}\right)\\ t>t_{st}:\;\;\;\;m_{dissolved}=m_{dissolved_{t_{st}}}+n\pi l\left(2r_{t_{st}}K_{0}^{II}\left(t-t_{st}\right)-\frac{K_{0}^{II\;2}}{\rho_{glass}}{\left(t-t_{st}\right)}^{2}\right) \end{array}$$
where $r_{t_{st}}$ and $m_{dissolved_{t_{st}}}$ are the fibre radius and lost mass after time $t_{st}$ when the steady-state Phase II is reached.

#### 2.2.5. Modelling Crack Growth and Strength Loss of Glass Fibres

Micromechanical models that predict the crack growth of GFs were proposed by Wierderhorn and Bolz in 1970 [132] for stress-corrosion and by Sekine and Beaumont in 1998 [155] for glass corrosion in acids. However, the link between dissolution chemistry-based kinetic models and the crack growth models was recently identified by Echtermeyer and Krauklis et al. in 2019 [145], even though it was initially proposed by Charles back in 1958 for bulk glass [156]. Echtermeyer and Krauklis et al. applied the DCZOK model as a chemical kinetics component to model the hydrolytic crack growth in GFs. In other words, by modelling the dissolution kinetics of GF, it is possible to predict the long-term deterioration of mechanical properties of the GFs due to hydrolytic crack growth [145].

Albeit in a slightly different context, a delayed fracture due to environmental attack on the fibre was attributed to a growth of the flaw back in 1913 by Inglis [157], in 1921 by Griffith [158] and in 1955 by Orowan [159] and further discussed by Charles in 1958 [156,160]. An analytical physics-based model was developed by Echtermeyer and Krauklis et al. in 2019 that quantitatively links the strength reduction in water to the chemical dissolution kinetics of glass ions migrating into the surrounding water, or in other words, to the increase in the Griffith flaw size of the fibres [145]. The model is based on the following fundamental concepts: the Griffith model for strength, a crack sharpness amplification factor linked to thermodynamics of surfaces and a zero-order (DCZOK) dissolution model. The rate of the increase is determined by the regular chemical dissolution kinetics of glass in water. Based on the corresponding dissolution constants, crack growth and strength reduction can be predicted for several water temperatures and pH. It was proposed that the crack-growth velocity should be related to the hydrolytic degradation rate of glass material [145] and that it should be possible to model the time evolution of GF strength loss using the DCZOK model (crack growth velocity should be related to a hydrolytic glass degradation rate) [139,141] if the kinetic dissolution constant is known, combined with a single crack sharpness amplification factor ϑ [145]. The strength reduction model reflects the two-phase behaviour and differentiates it from the previously developed empirical models [145].

Strength can be described by the Griffith equation [158], and an equivalent fracture mechanics expression is:(24)$${\hat{\sigma }}_{f}=\sqrt{\frac{2E\gamma }{\pi a}}=\frac{K_{Ic}}{Y\sqrt{\pi a}}$$
where ${\hat{\sigma }}_{f}$ is the strength; E is Young’s modulus, γ is the surface energy of the fibre, $K_{Ic}$ is the fracture toughness and a is the crack length. *Y* is a geometry correction factor for specimens of finite size; values for a rod with a crack can be found in [161]. 

Echtermeyer and Krauklis et al. hypothesized that the crack length returns to the initial crack length, and the crack velocity is obtained by the difference in crack growth and radius shrinkage of the fibre due to the dissolution of the glass in water [145]:(25)$$a\left(t\right)=a_{0}+\frac{\left(\vartheta -\xi_{sizing}\right)K_{0}}{\rho_{glass}}t$$
where a is the hydrolytic crack length after an ageing time t; $a_{0}$ is the initial crack length; ϑ is the crack sharpness amplification factor; $\rho_{glass}$ is the density of glass; $K_{0}$ is a dissolution rate constant; $\xi_{sizing}$ is the protective effect of sizing. 

The speed of the advancing crack is very slow–it takes around one minute to progress to the length of a chemical bond and yet it is sufficient to reduce the sized fibre´s strength by more than 30% within a month. 

Linking the strength degradation to the chemical dissolution kinetics at both degradation phases (dissolution constants $K_{0}^{I}$ and $K_{0}^{II}$) provides a quantitative link. Combining the model of the crack growth kinetics (explained in the previous module) with classic concepts of Griffith and fracture enables the prediction of GF strength with time. The time-dependent strength can also be expressed in relation to the static (t = 0) strength ${\hat{\sigma }}_{f0}$. The strength loss with ageing time can be expressed as in:(26)$$\{\begin{array}{c} t\leq t_{st}:\;\;\;\;{\hat{\sigma }}_{f}\left(t\right)=\frac{{\hat{\sigma }}_{f0}}{\sqrt{1+\frac{{K_{0}}^{I}\left(\vartheta -\xi_{sizing}\right)}{a_{0}\;\rho_{glass}}t}}\;\\ t>t_{st}:\;\;\;\;{\hat{\sigma }}_{f}\left(t\right)=\frac{{{\hat{\sigma }}_{f}}^{I}}{\sqrt{1+\frac{{K_{0}}^{II}\left(\vartheta -\xi_{sizing}\right)}{a_{0}\;\rho_{glass}}t}} \end{array}$$
where ${\hat{\sigma }}_{f}$ is the GF strength after an ageing time t; ${\hat{\sigma }}_{f0}$ is the static strength of GF; ${{\hat{\sigma }}_{f}}^{I}$ is the GF strength at the end of Phase I or the beginning of Phase II, in other words, i.e., ${\hat{\sigma }}_{f}\left(t_{st}\right)$; $t_{st}$ is the time when the steady-state Phase II is reached; $K_{0}^{I}$ and $K_{0}^{II}$ are the dissolution kinetic (rate) constants for Phase I and Phase II, respectively; $a_{0}$ is the initial crack length; ϑ is the crack sharpness amplification factor due to the thermodynamics; $\rho_{glass}$ is the density of glass; $\xi_{sizing}$ is the protective effect of sizing. 

The current limitation of the DCZOK model is the lack of a link from the fibres to the composites to predict the strength loss of FRPs. The effect of GF encapsulation in a GFRP is not fully understood [139]. In composites, the polymer matrix and sizing-rich interphase protect the glass and slow down the dissolution [137]. Quantitative results on how much precisely the matrix slows down the dissolution are lacking, except for one study for thin composite plates. It was found that even for composite plates with a thickness of a few millimeters, dissolution slows down by around two times when GFs are embedded in a polymer [137]. Additionally, it was found that fibre orientation affects the rate of glass dissolution. Dissolution degradation of GFRPs with fibres in hoop direction, rather than transverse, was the slowest [137]. 

Concerning ageing in other liquids, experimentally, GFs were reported to be inert in toluene and did not show any strength changes within the experimental error [145].

#### 2.2.6. Carbon Fibre Degradation

Surprisingly, there are not many articles that discuss the environmental stability of carbon fibres (CF). In a 2019 study, the environmental stability of SOFICAR TORAYCA T700SC 12000-50C carbon fibres was investigated [145]. CFs were reported to be chemically inert both in water and in toluene (one of the major oil compounds). Experimentally, CFs did not show any strength changes. However, it was reported that these CF bundles had a water- and toluene-soluble binder [145].

#### 2.2.7. Aramid Fibre Degradation

Aramid fibres and their derivatives such as trademarked Kevlar (Kevlar is the commercial name given by DuPont Inc., Wilmington, DE, USA), Twaron (Teijin Aramid), Nomex (DuPont), etc. are fibres made up of aromatic amides, i.e., poly(p-phenylene terephthalamide) or PPTA as in the case of Kevlar. Due to the presence of aromatic amide segments, these fibres have an inherent propensity for moisture uptake [162]. Such commonly found features further escalate this effect as microvoids and hydrophilic sodium salts on the fibre surface [162]. 

The reaction model for the Poly(m-phenylenediamineisophthal)amide (MPD-I) degradation was presented by Horta et al. in 2000 [163], followed by a publication describing the kinetic degradation model by Horta and Fernando V. Díez in 2003 [164]. They investigated the environmental degradation kinetics of various aramid fibres, such as Poly(m-phenylenediamineisophthal)amide (MPD-I) and poly(p-phenylenediamineterephthal)amide (PPD-T), which were among the essential high-tech polymers of the time [164].

The environmental degradation of aramid fibres can occur via three mechanisms: (1) oxidation, (2) hydrolysis, and (3) thermal decomposition [163]. The oxidation mechanism involves the transformation of the amine end groups (-NH_2_) into nitro groups (-NO_2_) [164]. The hydrolysis pathway undergoes splitting of the molecule into two fragments, causing an increase in amine (-NH_2_) and acid end groups (-COOH) [164]. The thermal decomposition mechanism produces the breaking of the polymeric chain into two fragments, increasing the number of amine end groups (-NH_2_). According to Yang, this reaction simultaneously produces random and specific thermal degradation lower molecular weight fragments such as MPD-I monomers, cyclic dimers, and cyclic trimers [165]. The degradation rate depends on the fiber polydispersity. The kinetic constants following all three possible mechanisms increase linearly as polydispersity increases [164].

While the works on molecular mechanisms and kinetics by Horta et al. are of special fundamental importance for modelling environmental degradation of aramid fibres, there are also other noteworthy publications. Chronologically, these include works by Shubha et al. (1993) [166], Connor et al. (1996) [167], Perry (1997) [168], Jain et al. (2000) [169], Lin et al. (2001) [170], Tanaka et al. (2002) [171], Menail et al. (2009) [172], Ramesh et al. (2013) [173], Menail et al. (2016) [174], Srivastav et al. (2017) [175], Engelbrecht-Wiggans et al. (2020) [176], Oğuz et al. (2021) [177], and Starkova et al. (2021) [66].

#### 2.2.8. Basalt Fibre Degradation

Concerning modelling of Basalt fibres, no models are available yet to the authors’ best knowledge. However, there are a few analogies that can be drawn from the natural ageing of geological basalt. Physical and chemical decay (degradation) affects natural basalt very quickly. In principle, it decomposes earlier than acidic igneous rocks (granite turns into sand plus layers of silicates). The “fate” of basalt minerals is smectite clay. A review of basalt fibre strength was shown in [178]. There have been several studies on the environmental ageing and stability of basalt and basalt fibres. Wei et al. (2011) looked into the degradation of basalt fibre/epoxy resin composites in seawater [179]. Sharma et al. (2014) investigated the degradation of basalt fibre–reinforced polymer bars in seawater and sea sand concrete environments [180]. Li et al. (2016) studied the temperature dependence of basalt weathering; basalt erosion was discussed [181]. Fan et al. (2021) investigated BFRP in humid and hot environments [182]. Glaskova-Kuzmina et al. (2021) studied the hydrothermal ageing of BFRPs [183].

#### 2.2.9. Natural Fibre Degradation

Natural fibres are mainly affected by water absorption (plasticization and swelling) and biodegradation with microorganisms [125,126].

### 2.3. Interphase Degradation Models

The composite interphase is formed from the multi-component sizing during the manufacture of FRPs [184]. This microconstituent has a proprietary composition. However, it is known that typical sizing consists of about five or so various chemicals [185,186]. Among them, coupling agents are the ones that have a critical role. They promote adhesion, stress transfer, interphase strength and hygrothermal resistance of the composite interphase [184,187,188]. The quality of the interfacial interaction is strongly dependent on the adhesional contact and the presence of flaws in the interphase [189]. The sizing-rich composite interphase degrades due to hydrolysis, forming the interphase flaws. These flaws may further develop into fibre/matrix debonding, matrix cracks and splitting along the fibres, as was observed by Krauklis et al. in [13]. The internal volume created by the flaws and cracks can be filled with water, leading to mass increase [13]. Based on the combination of experimental evidence of a year-long experiment by Krauklis et al. and ten-year-long test data by Perreux et al. (Davies’ group) [92], it was concluded that typical composites initially absorb extra water in the flaws and cracks created by interphase hydrolysis, such as shown in Figure 6 [13]. Eventually, these cracks will create a network that is connected to the surface of the composite laminate. When this network is formed, reaction products can leave the laminate, reducing the mass continuously [13]. 

There are no direct measurement methods to study the environmental ageing of the composite interphase. Some attempts have been made on quantifying the kinetics of sizing solubility in various fluids (styrene and acetone) [190]. However, when the interphase is concerned, it is still unknown how to quantify the interphase loss directly in the composite due to ageing. Furthermore, it has been noted by Riaño et al. that modelling techniques to study the composite interphase are becoming of high interest to the scientific community and industry [191]. 

Recently, a phenomenological mass balance approach was proposed for the hygrothermal ageing of GFRPs. It is based on the known ageing mechanisms of composite microconstituents (state-of-the-art phenomenological full representation of the interaction between the composite material and the water environment) [13]:(27)$$m_{interphase\;dissolution}\left(t\right)=m_{dry}+m_{water\;uptake}\left(t\right)+m_{oxidation}\left(t\right)-m_{leaching}\left(t\right)-m_{glass\;dissolution}\left(t\right)-m_{gravimetric}\left(t\right)$$

Mass balance enabled deduction of the kinetics of the hydrolytic degradation of the sizing-rich composite interphase, the rate of which was comparable to that of glass fibres’ dissolution in water [13]. However, the link between the mass loss and the composite interfacial strength loss is yet to be accomplished [4,13]. 

Joliff et al. worked on a numerical micromechanical approach to investigate the evolution of effective mechanical and interphase properties during the ageing of glass-fibre/epoxy composites [192]. 

Degradation of the interphase appears in significant differences between the directional diffusion coefficients that could not be explained by any structural micromechanical models [57,66,193]. As considered by the example of CFRP [73,193] and GFRP rods [66], high axial diffusivity (i.e., experimental values are highly overestimated compared to structural predictions) is likely to be related to a rapid diffusion path through a weak/debonded interface between fibres and matrix. An attempt to correlate the diffusivities and ratios with the durability performance of various FRP rebars was made in [66]. By analyzing the data of up to 15 years long moisture diffusion and mechanical short-beam tests, the ratio of the directional diffusivities of composites was reasonably related to the ILSS reduction.

## 3. Modular and Multiscale Approaches

### 3.1. Modular Approach

Modular modelling approaches for composite environmental ageing was proposed by Echtermeyer et al., first presented in [2]. The modular approach was further expanded in (a step towards the multiscale paradigm of the composite ageing) [4,139]. The modular approach described therein is schematically shown in Figure 7. 

### 3.2. Multiscale Simulation Frameworks

Since ageing phenomena are often complex to model and composites are multiscale materials in nature, the modular approach described in the previous section, separately describing degradation mechanisms for each microscopic material constituent, comes as a natural choice. However, the spatial scale of interest for structural design is often several orders of magnitude larger than the characteristic sizes of microscopic components being described. When modelling ageing in practice, it, therefore, becomes necessary to combine and upscale microscopic phenomena up to a larger scale of interest, a common goal that pervades every multiscale ageing framework we review in the following.

Multiscale modelling can be performed in several different ways, each coming with its own set of advantages and drawbacks. In the following, we provide a brief recap of these strategies and illustrate them in the context of works dealing with the ageing of composite materials and focusing on works that employ Finite Element (FE) simulations on at least one scale of interest. A condensed list with recent works on the subject is also presented in Table 2.

### 3.3. Direct Numerical Simulation (DNS)

A straightforward approach to reaching higher scales is to simply model the microstructure explicitly throughout the complete macroscopic domain of interest, a strategy known in the literature as Direct Numerical Simulation (DNS). Although being the approach with the highest possible modelling fidelity, DNS comes with the drawback of being extremely computationally expensive and is usually employed to verify the predictions of computationally cheaper multiscale strategies.

Examples of DNS being used in the context of ageing of composites can be found in the series of works by Joliff et al. [194,195,196]. Here, the authors use the approach to study the influence of microscopic fibre arrangement on diffusion kinetics [194,195] and swelling stress distribution [196] of unidirectional GFRPs. Although computationally expensive, a DNS model of an accurate microstructure coming directly from microscopic observations enabled the authors to postulate the hypothesis that the interphase region around the fibres has a higher diffusivity than the bulk resin. Other recent examples of DNS being used to model ageing can be found in [104,197].

### 3.4. Analytical Homogenization (AH)

This approach consists of deriving analytical expressions for combining properties coming from different materials at a lower scale towards a homogeneous higher-scale representation. Popular strategies include rule-of-mixture approaches [213,214] and more complex mean-field homogenization techniques [215,216,217]. These models are usually limited in their ability to represent general geometries or consistently combine highly nonlinear phenomena but are computationally very efficient.

In [198], the authors propose a new rule-of-mixtures approach for directional diffusivity of plain-woven composites, which considers resin-rich gaps between fibre tows. The outcome is a fast approximation of the diffusivity coefficients that agrees well with high-fidelity FEM models. Krauklis et al. [112] propose an analytical model for computing directional swelling coefficients in fibre-reinforced composites with good agreement with micromechanical FEM models. The authors in [199] use the Halpin-Tsai method and macroscale FEM simulations to elucidate the mechanisms behind the ageing of GFRPs under sulfuric acid exposure. These strategies are also successfully employed in the context of ageing of composite materials by several other authors, for instance [200,201].

### 3.5. Numerical Homogenization (NH)

In contrast to the analytical approaches, numerical homogenization consists of obtaining averaged properties from a numerical model of the material microstructure (using, e.g., FEM). In order to avoid the extreme computational cost of DNS, the goal here is to define a Representative Volume Element (RVE), also called Unit Cell (UC) or Representative Unit Cell (RUC)—which is small enough to limit computational effort and large enough to be representative—solve it for several scenarios and use quantities averaged from these solutions to calibrate a pre-defined constitutive model at the macroscale. Figure 8 shows a few examples of the RVEs used by different authors to obtain homogenized properties for fibre-reinforced composites. This is a popular approach capable of upscaling nonlinear behaviour from arbitrarily complex microstructures as long as a suitable higher-scale model can be defined.

Numerical homogenization is a widely-used technique in modelling ageing. Rocha et al. [202] employed the technique to obtain insight on the directional diffusivity behaviour of glass-epoxy composites. As is customary with the method, periodic boundary conditions are applied to the RVE in order for the homogenization operators to retain consistency. The homogenized diffusivity coefficients are then computed by applying concentration gradients in each direction and computing the volumetric average of the resulting flux. A similar strategy was used by Krauklis et al. for computing directional swelling coefficients [112]. In [200], aside from proposing an analytical solution for diffusivity in highly filled polymers, the authors also perform an extensive numerical homogenization study in which the accurate representation of the microstructure afforded by the microscopic FEM model helps elucidate the role of tortuosity on the homogenized diffusivity. Other studies use numerical homogenization for an interface damage profile after hygrothermal degradation [203], diffusivity in woven composites [204] and degradation through a combination of UV and condensation [205,206].

### 3.6. Computational Homogenization (CH)

The appeal of numerical homogenization lies in the fact that once a suitable macroscopic model is calibrated, no additional microscale simulations are necessary. However, the combination of synergistic microscopic ageing mechanisms is often too complex to be fit by a single pre-calibrated macroscopic constitutive model. In Computational Homogenization (CH)—also known as *Concurrent Multiscale* or *FE*^2^—the idea is to concurrently model two or more scales without making any constitutive assumptions during upscaling [218,219]. This is achieved through a continuous scale link enforced by embedding an independent RVE model at each integration point of the macroscopic domain and solving the associated microscopic boundary-value problem every time the macroscopic constitutive response needs to be computed.

Computational homogenization is an emerging trend in high-fidelity solid mechanics and is currently being used and extended by several research groups, including in applications to complex material failure [220] and multiphysics thermomechanical [221,222,223,224] and chemo-mechanical [225] problems. Studies that explore CH for ageing are significantly scarcer. Among the earliest contributions is the framework proposed by Terada and Kurumatani [207], including chemical degradation, micro-crack formation and propagation and couplings between diffusion and stresses (swelling/contraction) and damage and diffusivity (through micro-cracking). Along the same lines, Bailakanavar et al. [208] propose a CH framework combining macroscopic moisture diffusion with microscopic swelling and material degradation.

In a more recent series of articles [202,209,210], Rocha et al. propose a framework specifically for hygrothermal ageing of fibre-reinforced composites (Figure 9). At the macroscale, a transient diffusion model is combined with a stress model in a staggered way (weak coupling). At the microscale, swelling, resin plasticization and interfacial degradation are modelled as a function of moisture concentration and combined with cohesive-zone models for the interface and a pressure-dependent viscoelastic–viscoplastic model with damage for the resin. The concurrent scale coupling allows for macroscopic transient swelling stresses to influence microscopic failure.

### 3.7. Multiple Time Scales and Other Approaches

The list of different multiscale approaches presented in the previous sections is by no means exhaustive, and several other strategies for upscaling material behaviour have been employed by researchers throughout the years. Oskay [211] proposes a framework for fully coupled diffusion mechanics, including concentration-dependent material properties. The adopted multiscale approach is a variational formulation of asymptotic homogenization that limits computational effort through a domain decomposition strategy that only resolves the fine-scale at selected points of the macroscopic domain. Shi et al. [212] also opt for a hybrid approach, combining DNS at regions of interest with homogenized macroscopic models calibrated through numerical homogenization at the rest of the domain. The result is a flexible high-fidelity simulation framework for braided composites subjected to thermo-oxidative ageing.

The preceding discussion has focused on simulating ageing effects in two or more spatial scales at the same time. However, given that ageing often comprises a combination of phenomena acting at drastically different time scales, e.g., the combination of fast cyclic mechanical loading with slow moisture diffusion, performing multiscale simulations in time also becomes relevant. Although still not extended to treat the specific challenges of ageing of composites, a number of authors employ the strategy on multiscale and multiphysics problems. In the seminal work by Yu and Fish [226], the authors propose an asymptotic homogenization framework that treats multiple spatial and time scales using the same homogenization principles and employs it to solve coupled thermo-viscoelastic problems. Haouala and Doghri [227,228] extend the approach to viscoplasticity, allowing for its use in more complex material models. Alternative time homogenization approaches [229] and modelling techniques such as the LATIN method [230] could also be employed to build time-homogenized models for ageing.

### 3.8. Accelerating Multiscale Simulations

High-fidelity numerical simulations of material behaviour require significant computational efforts. When performing multiscale simulations for which no suitable higher-scale model is available, this cost increases significantly as opting for Computational Homogenization becomes necessary and microscopic simulations must be performed at every integration point of the higher-scale model. Worse yet, modelling ageing often requires a large number of such simulations, for instance:When performing long-term predictions of material degradation under cyclic environmental exposure, with transient simulations requiring a large number of time steps;As part of many-query applications such as design optimization, in which evaluating each trial design requires a complete set of high-fidelity simulations to be run;When employing models as part of Structural Health Monitoring (SHM) frameworks requiring inverse problems to be solved on the fly as new sensor measurements are obtained.

This essentially renders concurrent multiscale modelling unfeasible for many practical applications. Thus, there is an urgent need to accelerate the micromechanical simulations that represent the main computational bottleneck of multiscale approaches. In this section, a brief overview of techniques used for improving the efficiency of multiscale modelling of ageing are provided.

### 3.9. Model Order Reduction (MOR)

Model Order Reduction (MOR) can refers to a wide range of acceleration techniques that have the general aim of reducing the number of degrees of freedom of the boundary-value problem being solved. The term is often employed to refer to projection-based reduction, consisting in projecting the original high-dimensional problem onto a lower-dimensional subspace spanned by representative solution modes on which fast approximate solutions can be found [231,232,233]. Such models are usually combined with a second reduction stage that significantly reduces the number of integration point evaluations needed to solve the model (the so-called hyper-reduction stage [234,235,236]). Rocha et al. [210] use subspace projection to accelerate multiscale simulations of hygrothermal ageing in composites. With a combination of the Proper Orthogonal Decomposition (POD) and Empirical Cubature Method (ECM) techniques, reduced models were built approximately 300 times faster than their full-order counterparts, allowing for an extensive set of multiscale ageing simulations to be performed.

Another approach to construct reduced models that have seen use for modelling ageing consists of combining Nonuniform Transformation Field Analysis (TFA) with asymptotic homogenization [237]. In [208], it has been used to build a multiscale/multiphysics modelling framework for coupled diffusion-stress simulations in polymer matrix composites, including swelling and concentration-dependent material properties. In [238], the approach is further extended to a diffusion-reaction scheme to model oxygen diffusion, oxidation and deformation in ceramic matrix composites.

### 3.10. Machine Learning (ML) Approaches

The primary motivation for to the use of Computational Homogenization is the lack of a suitable mesoscale model that are currently able to accurately capture the highly nonlinear behaviour arising from interactions between ageing mechanisms at the microscale. An alternative that is currently rising in popularity is to abandon the search for a physics-based mesomodel and instead resort to purely data-driven surrogate models built with machine learning techniques. Several machine learning models for regression (e.g., neural networks, Gaussian processes) are universal approximators, i.e., capable of approximating any continuum function up to an arbitrary level of precision [239] and therefore reproduce arbitrarily complex nonlinear behaviour.

Although increasingly popular in building surrogate models for stress behaviour in order to accelerate Computational Homogenization [240,241,242], studies that employ the approach in combination with multiscale simulations of ageing are still rare. Nevertheless, machine learning is being extensively used both to fit experimental degradation data (regression) as well as to detect defects in a condition monitoring setting (classification) [243].

Doblies et al. [244] used Artificial Neural Networks (ANN) to predict the degradation of epoxy resins exposed to oxidation and thermal ageing. The trained model was capable of accurately predicting exposure time, temperature and residual strength based on input FTIR spectra. Valenzuela et al. [245] built an ANN model to predict water uptake across a wide range of biodegradable polymers using several different material descriptors as input with promising results. Nguyen et al. [246] employ a set of Finite Element simulations in order to train a Recurrent Neural Network (RNN) model for the prediction of hygrothermal ageing in laminated composite plates. The hidden, latent representation learned by the RNN allows for time-dependent behaviour to be successfully predicted.

Several works also focus on the Bayesian machine learning for building robust predictive models, the appeal being not only the ability to provide predictions for inputs not seen during training, but also the provision of a measure for the uncertainty associated with these predictions. Bayesian inference also allows for a principled way to obtain insight on unobserved model variables and perform inverse modelling in a robust way that is less sensitive to overfitting in low-data scenarios. Keprate et al. [247,248] construct Bayesian networks to predict the durability of composite materials under environmental degradation by combining multiple degradation mechanisms. Shabouei et al. [249] employed an extensive combination of Bayesian machine learning techniques in order to predict complex diffusion-reaction phenomena in Ni/Al composites across the scales, including Bayesian inverse modelling with Polynomial Chaos (PC) surrogates and multi-fidelity Gaussian Process (GP) emulators for efficient computation of quantities of interest.

## 4. Emerging Trends in Degradation Modelling of Biodegradable Polymers

### 4.1. Biodegradation—An Introduction to Terms and Definitions

Plastics can be divided into four characteristic groups. A two-axis model has been used most often in the literature. The vertical axis shows the biodegradability of plastics, whereas the horizontal axis represents the material origin (petrochemical or renewable raw materials). In addition to non-biodegradable plastics made from petrochemical raw materials (classic or traditional plastics), biodegradable plastics are made from renewable raw materials. These are plastics that have been produced from biomass-containing raw materials and are biodegradable. Furthermore, biodegradable plastics that can be biodegraded were produced from fossil raw materials. Finally, there are the non-biodegradable plastics made from renewable raw materials produced from biomass but are not biodegradable [250]. According to an ASTM definition, biodegradable polymers and plastics are materials that can be quantitatively converted to either carbon dioxide and water or methane and water under aerobic and anaerobic conditions [251].

Biodegradable polymers (e.g., polyesters, polyamines) have a higher concentration of heteroatoms than fossil-based polymers with pure carbon backbones, such as polyethylene. Furthermore, biodegradable polymers are often derived from raw materials inferior to their petroleum-based counterparts [24].

There are usually three stages (fragmentation, hydrolysis and assimilation) that are part of the biodegradation process in enzymatic hydrolysis. Microbial secretases catalyse this process. In the first step, small pieces (microplastic particles) are formed by weathering, UV-irradiation, mechanical forces and microorganisms. Subsequent hydrolysis at the ester bond of the polymer results in the reduction of molecular mass and the formation of soluble monomers and oligomers. These breakdown products are taken up by intracellular enzymes and used as a carbon and energy source to produce a greater cell biomass and end products such as carbon dioxide and water. This is called the assimilation process (see Figure 10, [18]).

The hydrolysis of the polymer can be either biotic or abiotic, with abiotic hydrolysis being much slower than biotic or enzymatic hydrolysis [252,253]. In a natural environment, both biotic and abiotic factors synergistically influence biodegradable polymers with respect to compositions and processes [24,62,254]. Several aspects influence the microenvironment, such as pH, redox potential, interfacial chemistry and physics, chemical composition, crystallinity, and polydispersity [24]. Figure 11 left shows the interaction of abiotic and biotic factors and the interplay of microorganisms such as bacteria, fungi, archaea and algae [255], environment and the polymer as a complex multivariable space of biodegradation. The movement of the polymer chains becomes restricted with increased crystallisation. As a result, fewer polymer chains are available for degradation products such as microbial lipases or other ester-cleaving molecules. Previous studies have shown that the degradation of PHBV (poly(hydroxybutyrate-co-valerate)), catalysed by lipases, occurs preferentially in the amorphous regions [256]. As shown on the right in Figure 11, the microbial enzymes are thus only able to degrade the amorphous regions of the polymer. The adjacent crystalline regions limit the depth of degradation. Essential factors and chemical groups impaired during the biodegradation process are shown at the bottom of Figure 11 [257]. Hydrophilic microbes usually have higher degradation rates than hydrophobic microbes, as was demonstrated in the literature [258].

### 4.2. Data-Driven Approach to Elucidate Degradation Trends

Min et al. addressed the question as to how many descriptors would be needed to model the degradation behaviour of plastics in the ocean or to understand if degradation is possible. They present a so-called data-driven approach to investigate the degradation trends of plastic debris. The biotic and abiotic degradation behaviour in seawater is linked to physical properties and molecular structures [62]. Figure 12 exhibits the flow chart to calculate the hydrophobicity. It is based on a molecular level method combining theory, simulation, and experimental validation [260]. 

Here, negative Log*P* values predict water solubility, for polymers that swell in water or polymers that tend to absorb moisture. Positive values, on the other hand, predict insolubility. Different polymers can be compared by using MD (molecular dynamics simulation) to minimise the energy of molecular models and then calculating the surface area (SA) [62].

The wide range of hydrophobicity of plastics is revealed in Figure 4 [62]. Water-soluble plastics (see blue colour) such as PEG (polyethylene glycol) or PVA (polyvinyl alcohol) exhibit polar groups such as OH groups degrading via microbial oxidation [261]. In the second category (yellow colour), the erosion tendency of the polyester surface correlates with hydrophobicity when T_g_ values are <T_Ocean_. PA (polyamide/Nylon) is the most used fishing gear material belonging to the yellow group (Nylon 66 and Nylon 6). Nylon is very persistent in the oceans despite the possible fragmentation, having a very long lifetime and contributing to a major extent to e.g. ghost fishing and plastic debris [262].

The third group (see red) includes very hydrophobic plastics with no functional groups for abiotic hydrolysis but with a large percentage of C-H bonds susceptible to photodegradation. For PE and PP, extremely slow surface erosion can be observed (in addition to oxidation by photoinitiated processes). PE and PP have been identified as polymers predominantly at or near the sea surface [263]. It is interesting to note that the Log*P*(SA)^−1^ values for these very hydrophobic plastics show lower densities, which allows them to float near the sea surface. The ranking is shown in Figure 4, and tends to correlate with the propensity for polyester degradation. However, plastics with T_g_ values > T_Ocean_, including polymers such as PLA, PLLA and PET, degrade more slowly than expected. The degradation of PLA in seawater is very slow, in contrast to degradation under composting conditions [264]; see also Table 3 in [18]. This shows that multiple metrics are required to understand the degradation of polymers in the ocean. Therefore, crystallinity, enthalpy of fusion, Tg, molecular weight, and Log*P*(SA)^−1^ values were studied in pairs to find patterns of degradation. Figure 13 compares crystallinity and enthalpy of melting with values of Log*P*(SA)^−1^ for both biotic and abiotic conditions. Surface erosion was calculated using the surface area of each plastic (SAbulk), mass loss and the number of days in ocean water. To obtain the systematic diversity of hydrophobicity values, the number of -CH_2_- units in the monomer structures was calculated between 5 for PPS and 11 for PPSeb. As a result, Min et al. recently demonstrated that the enzymatic degradation of polyesters with T_g_ values below the ocean temperature is faster than for abiotic hydrolysis. In the case of abiotic hydrolysis, biodegradation and photoinitiated processes co-occur. It is likely that the decrease in molecular weight due to abiotic hydrolysis or photoinitiated reactions could facilitate biotically induced processes, whereas enzymatic hydrolysis could promote abiotic hydrolysis. It appears that abiotic hydrolysis (see Figure 13a,c) is probably more sensitive to increases in hydrophobicity, enthalpy of melting, and crystallinity compared to biotic processes. Biotic processes exhibit faster rates in more hydrophobic polyesters, such as PPPim) and PPSub. Moreover, the comparison of polyesters and PA (e.g., Nylon 6 and Nylon 6,6) demonstrates that biotic and abiotic processes also occur in semi-crystalline plastics, but crystallinity slows down these processes. More details are provided in [62].

In their recent study, Yamawaki et al. performed virtual experiments for compost degradable polymers. As stated, no efficient system has yet been developed for practical use that considers multiple decomposition factors. Yamawaki et al. developed a prediction model for the degree of degradation, so as to investigate the degradation factors based on analytical data and experimental conditions [265]. This involved creating a predictive model using machine learning on a data set. The molecular weight received by GPC was the objective variable; the explanatory variables were the moisture content in a compost environment, the degradation time, the degree of crystallinity before the experiment and the NMR spectra. A decomposition degree predictive model was developed by considering the weight loss ratio as the objective variable. Different machine learning algorithms have compared the predictive accuracy of the predictive models. Furthermore, the NMR spectra have been compressed dimensionally using PCA (Principal Component Analysis). Explanatory variables were taken from the NMR data (crystallinity as analytical data and moisture content in the compost environment as experimental conditions). Figure 14 shows as a summary the concept of real and virtual degradable experiments that have been used. 

Decomposition experiments were conducted to achieve the ideal experimental conditions and different analytical data values (see Figure 14 left). The optimal degree of decomposition, different analytical values and experimental conditions were investigated by virtual experiments in combination with a predictive model and a Bayesian optimisation method (see Figure 14 right). Statistical variables were used to confirm the good accuracy of this predictive model. However, additional experimental conditions such as temperature and pH could lead to a more accurate predictive model [265].

The applicability of TTSP to biodegradable polymers has been proved in [266,267,268]. Amiri et al. considered the biobased polymer a thermorheologically complex material and implemented vertical shifts to obtain smooth master curves [268].

### 4.3. Laboratory vs. Field Experiments: Outlook

Lott et al. recently investigated the performance of biodegradable plastic materials under marine-environment conditions. Biodegradation lab tests were compared with field tests. Furthermore, statistical modelling has been used to predict the half-life for the investigated materials under different environmental conditions [23]. The results have shown high variability in the biodegradation rate of biodegradable plastics under marine conditions, and the specific half-life times differed by different orders of magnitude ranging from weeks to years. Earlier, Albertsson and Karlsson, for instance, investigated PE photo-degradation in a so-called inert system in a 10-year experiment. They demonstrated that the PE degradation rate can be characterized by three different steps [269]. The biodegradation exhibits a complex phenomenon. It is still challenging to realize nature-like experiments in the laboratory. Many parameters must be taken into consideration. Each biodegradation stage needs an adapted estimation technique [254]. Biodegradation tests under laboratory conditions could lead to an overestimation of the biodegradation level regarding biodegradable plastics. Therefore, test standards should be developed for biodegradation to reduce the gaps between natural biodegradation compared to investigations carried out in the lab [270]. It has been shown previously in a 1000-h accelerated ageing test simulating outdoor conditions that biodegradable gillnets degraded faster than the nylon, which was used as a reference gillnet [31]. Based on these lab experiments, well-tailored biodegradable polymers and PA for fishing nets (gillnets) will be investigated in a long-term trial. Different laboratory tests and field trials will be carried out to ensure that the results represent the environmental factors prevailing in other countries [271]. Additionally, statistical models and data-driven techniques, and importantly, the investigation of mechanical forces on the rate of degradation, which needs further research, will help better understand the degradation behaviour of polymers in the sea and under composting conditions in the future.

## 5. Economic Role of Degradation Modelling

The economic role is versatile as polymers range from negligible value items for daily packaging to the most complex technologies and are valuable for the aerospace industry, mainly determining the costs for testing that makes significant part for the development of new products. Since, for development of the new materials, validation is expensive and time-consuming, the bottleneck is time and funding—modelling might be the way to replace testing programs, which would be beneficial for providing new innovative materials faster to the market. Polymers are tested for physical and chemical properties and their degradation, all of which require resources. Often there is a requirement that polymers have durability properties for an extended period; however, another aspect is that they need to have degradation properties after the life cycle has been finished to guarantee a lower environmental burden that can also be calculated in an economic sense. Predicting the influence of factors and their combinations is complex. Each polymer on top of it has distinct rules on how the product shall be used to avoid unexpected impact through the instructions of use. Despite the durability properties, testing and development of equipment in industrial laboratories, predicting environmental ageing is still complex. Standard accelerated ageing tests are approximate tools compared to actual exposure conditions, coupled environmental influence and corresponding ageing mechanisms.

Global plastic production has grown almost exponentially and reached 390 Mt in 2018. It generates 250 Mt of waste per year, from which 175 Mt are collected; for 75 Mt, the destiny is unknown. Recycling is performed for 50 Mt, energy recovery achieved for 50 Mt, and managed landfills accept 75 Mt, improper disposal and leakage are estimated at 62 Mt and 13 Mt, respectively [272]. From the environmental point of view, the latter two positions are the most essential–improper disposal and leakage. If inappropriate disposal in ideal conditions is fixed, the leakage part might never be controlled. Leakage also includes the degradation of polymer products, and accurate modelling might act as a necessary tool to avoid environmental costs linked to the issue as so far the landfilling and incineration have led already to too serious contamination consequences [259,273]. Another way to prevent this is by substituting polymer products and using fewer polymers for low durability items that decompose too fast and are not reusable/recyclable. Regarding lifetime, plastic products are short-life (e.g., packaging), middle-life (e.g., for agriculture, electronics) and long-life, as for building and construction [274].

On the other hand, testing laboratories are working on testing efficient degradability, so the product after the life cycle ends becomes less damaging to the environment [275]. Recycling and recovery are not a priority for the circularity approach. The basic solutions include the diminishment of consumption and avoiding short-lived products is the best solution for the future. Testing and modelling organizations are challenged with estimating the physical, chemical and industrial properties and the opportunities for substitution with other less environmentally damaging materials [276]. “Plastics 2030” is an EU plan aiming to avoid the leakage of plastic into the environment and promote reuse and recycling [275]. Proper modelling of polymers’ ageing is the way how to prevent the production of improper plastic products and diminish the leakage per se. The polymer industry has been involved in the global Declaration for Marine Litter Solutions; it is an alliance of 74 plastics associations and rides over 350 projects to combat this problem.

The calculation of the environmental impacts of plastic and alternative materials into monetary values is possible by Trucost’s Natural Capital Valuation techniques. The techniques help to estimate the value of environmental goods or services if there is no specific market price. It applies to water abstraction, land and water, air pollutants, and emissions of carbon dioxide as well as marine litter. The environmental cost overall is largely attributed to the amount of marine litter in most assumptions. Other environmental impacts are region-specific–packaging ends up in the same year as waste. Still, durable products are assumed to have a lifespan of multiple years (depending on the product) and are discounted for the selected calculation year [277,278].

One method of calculation is to link biodiversity, measured species richness, net primary productivity (NPP) [279], and ecosystem service value (ESV). Percentage change in ESV per unit emission of pollutant at the country and substance level and applied this percentage to the average value of one square meter of the natural ecosystem in each region globally, in addition to sensitivity analysis, economic and biophysical modelling.

The total environmental cost of plastic-use in the consumer goods sector is >$75 billion a year, which includes production and the cost of their impact on the ocean. Initial analysis of the transport of plastic and alternative materials includes adjustments giving $139 billion per annum [278].

Estimation of durability’s impact on the environment is achieved using a life cycle analysis (LCA) [280,281]. Various categories are used to quantify the ecological effects of composite durability. Later, it is possible to compare LCAs from a composite material in a comprehensive analysis while taking into account the durability aspect or without it. Environmental ageing itself is weathering similarly to nature. Models take artificial weathering’s regularity of different natural cycles, intensity, duration, exposure conditions; however, these may not precisely depict natural conditions, and modelling adjustments shall be performed [282,283]. With this arises the difference between so-called green composites and synthetic composites; for the latter, the modelling may be better standardized for mass production; on the other hand, the environmental impact of synthetics is more severe.

Predictions are one of the best features for a kinetic analysis. They are used to estimate the kinetic behaviour of materials beyond the experimental measurements [16]. The discussed models and other phenomenological models (covered in Part 2 of the Review [12], such as Arrhenius and Williams–Landel–Ferry (WLF) are used to predict the durability of the polymer product, thus saving on costs of direct testing.

## 6. Discussion

Substantial gains and savings of resources such as time and money can be achieved through the use of modelling and simulation to understand system performance. Modelling and simulations are tools that use ample amounts of data, machine learning to diminish and completely replace the need for direct testing, improve cost and operational effectiveness analyses, and strengthen system development. Polymer ageing modelling would encourage the wider use of improved properties materials in the industry and safety, cost, and environmental advantages over operational testing. Augmented virtual operational test design and evaluation for simulation of ageing processes should include: (1) a fully comprehensive description of theoretical assumptions, (2) validation, (3) uncertainty analysis, (4) sensitivity analysis. The simulation requires a repository of information about a system’s past and current performance, using relevant sources of information, systems similar to machine learning, the promotion of early operational assessments and testing, and extrapolate models.

Physical models require pre-existing models before they can test for failures. Finite Element Analysis (FEA) uses mathematics to model the physical world’s influence on the design process. A large proportion of the cost for polymer development is determined by the decisions made early in the design process, which incorporates the testing phase. Implementing FEA reduces the number of prototypes needed because it can perform on-the-fly testing and product validation in a virtual space. Testing is the most time-consuming part of product development, and when failures occur, changes have to be implemented. Modelling and simulations would significantly reduce these costs.

As mentioned in the above chapters, modelling and simulations might significantly decrease costs related to product development. They will reduce the environmental cost, the costs associated with a sustainable circular economy, as will the cost an enterprise submits in their Income Statements. With that being said, if the cost of one particular material development is, for instance, USD 200,000, this amount would extend multiple times if total environmental costs are to be considered. Therefore, the total costs might rise to >USD 1,000,000 depending on the calculation methodology. Moreover, existing legislation proves to be more environmentally aware. It will force to use less natural capital for experiments and developments and replace that with high taxes, which is primarily done in many countries for the basic resources extraction and through pollution permits control system.

In Figure 15, the authors generalize the way this innovative modelling approach, through its stages, in concert with a Cost–benefit Analysis, can indicate the quantitative indicators of performance and SWOT, which could provide the qualitative aspect for the synthesis of a decision system for product development.

Future sustainability approaches are required to consider ESG (environmental social governance) aspects in decision making for product development and modelling, and a significantly reduced impact on the environment is ideal. It will provide lower costs, less environmental impact; however, we should also consider the safety aspects. The modelling should provide product safety, although moving away from physical testing entirely in material development is not possible; instead, we should seek to replace these aspects as much as possible.

## 7. Conclusions

Service lifetimes of polymers and polymer composites are impacted by degradation processes occurring during environmental ageing. State-of-the-art multiscale and modular approaches of predicting environmental ageing of polymeric and composite materials have been systematized and discussed, with this study providing a comprehensive overview of the broad possibilities and present limitations of the currently available models.

In light of the increasing interest in biopolymers, a “two-edged sword” nature of the ageing phenomena was discussed. We have identified applications or which degradation is favourable (biodegradable plastics) and those for which it is unfavourable (structural polymers and composites). In conventional degradation, the objective is to retain the material within the useful lifetime, whereas, in biodegradation, the end of life is simulated. Therefore, the same ageing phenomena and modelling approaches would be applicable if the “decomposition” criteria replace the “safety” criteria. 

For more conventional structural polymers and composites, the modelling approaches to predict environmental ageing have been shown to be a more affordable alternative to costly testing programmes. They are expected to partially replace the testing procedures, reducing the validation costs and lowering the “bottleneck” for new composite projects. Some accelerated testing methodologies and phenomenological models for predicting the durability of polymers and composites will be highlighted in the next study.

## Figures and Tables

**Figure 1 polymers-14-00216-f001:**
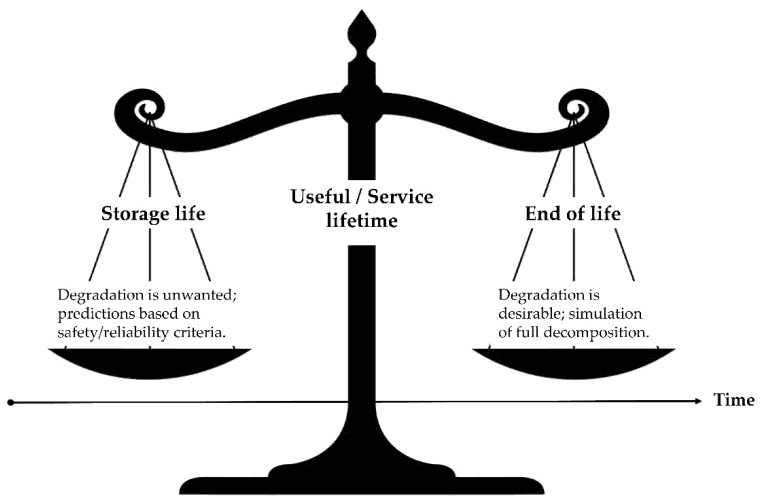
Time scale and critical points for materials’ life show the “two-edged sword” nature of the degradation processes (*clipart is attributed to “Scale Clipart Transparent—Creative Commons Clipart”*).

**Figure 2 polymers-14-00216-f002:**
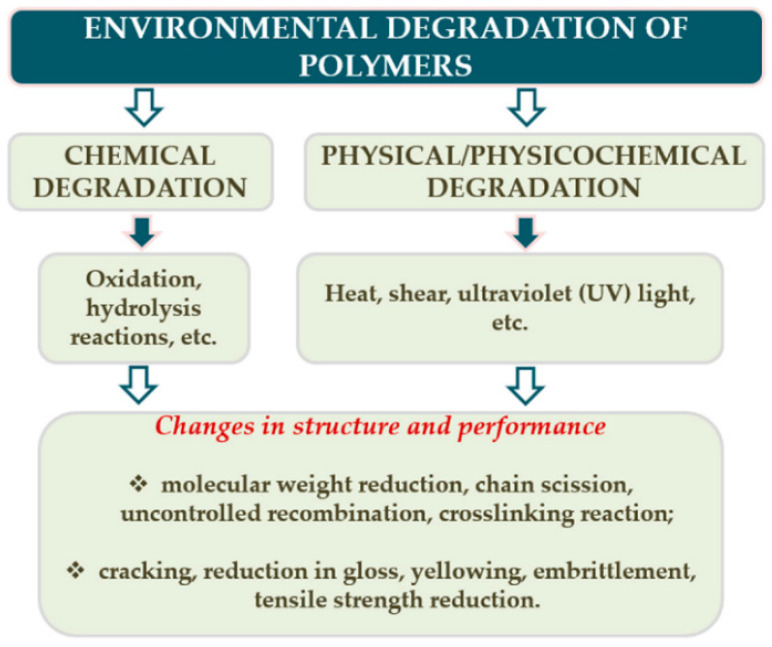
Schematic representation of the environmental degradation of polymers: chemical and physicochemical degradation factors [16].

**Figure 3 polymers-14-00216-f003:**
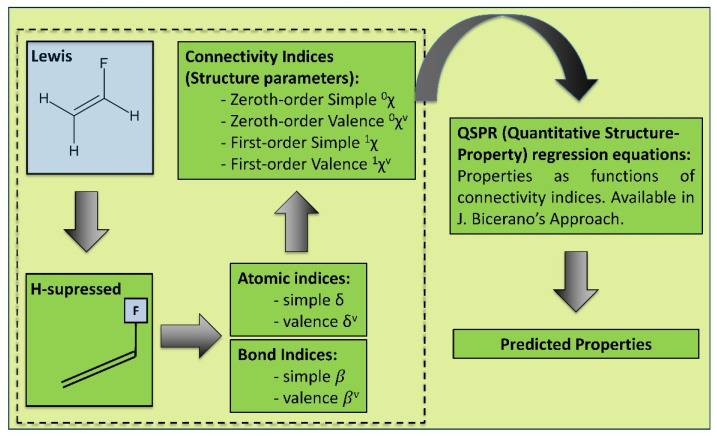
Quantitative Structure-Property Relationship (QSPR) approach; topological method and connectivity indices [38], reproduced from [4].

**Figure 4 polymers-14-00216-f004:**
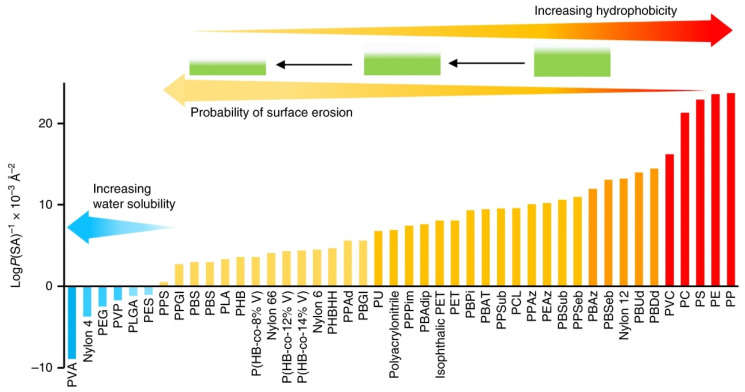
The wide range of hydrophobicity of plastics [62].

**Figure 5 polymers-14-00216-f005:**
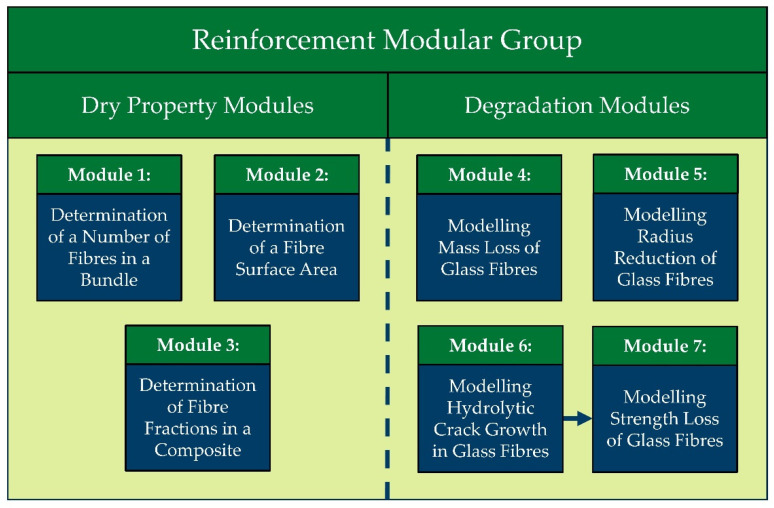
The seven developed modules of the Reinforcement Modular Group within the Modular Paradigm reproduced from [139].

**Figure 6 polymers-14-00216-f006:**
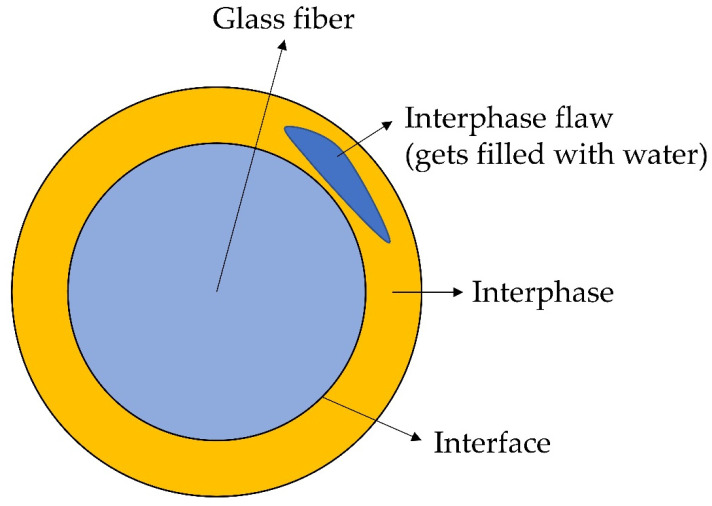
The interphase flaw is formed and gets filled with water, reproduced from [13].

**Figure 7 polymers-14-00216-f007:**
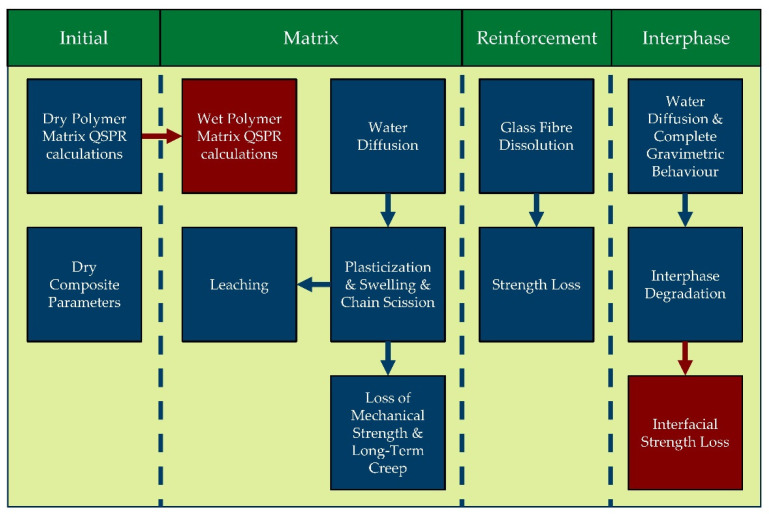
Modelling toolbox modules for predicting environmental ageing of composites. Red coloured modules indicate that such models are not yet available, whereas blue ones are complete (and described in the previous chapter).

**Figure 8 polymers-14-00216-f008:**
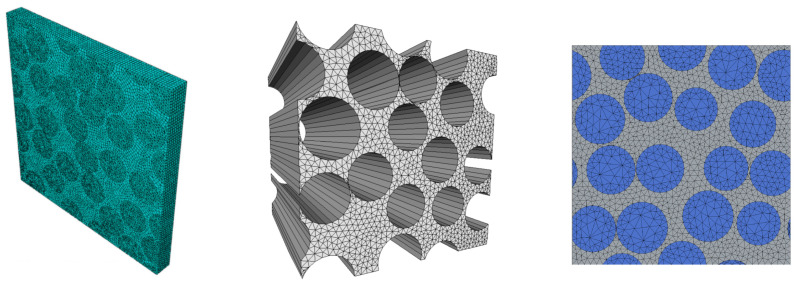
Examples of Representative Volume Elements (RVE) used in different studies [112,202,210].

**Figure 9 polymers-14-00216-f009:**
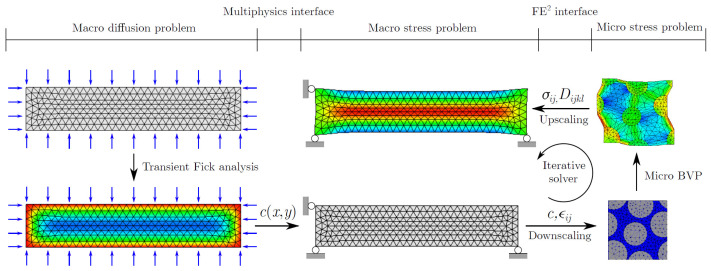
The FE2 framework for hygrothermal ageing was proposed by Rocha et al. [209].

**Figure 10 polymers-14-00216-f010:**
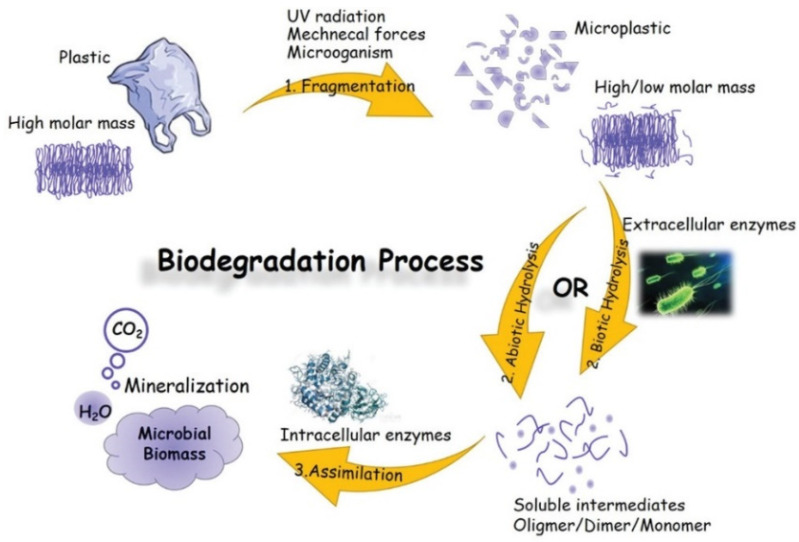
The different steps are involved in the biodegradation process [18].

**Figure 11 polymers-14-00216-f011:**
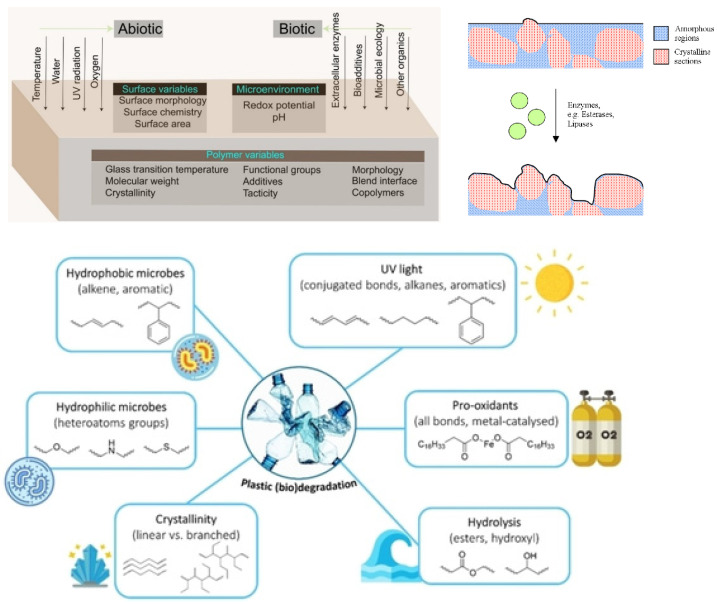
**Left**: Interplay of biotic and abiotic factors [24]; **Right**: The effect of polymer crystallinity on enzymatic degradation [259]. **Bottom**: Important factors and chemical groups impaired during biodegradation [257].

**Figure 12 polymers-14-00216-f012:**
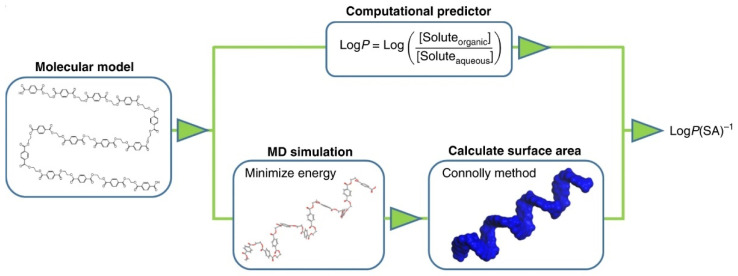
Flow chart for calculating hydrophobicity [62].

**Figure 13 polymers-14-00216-f013:**
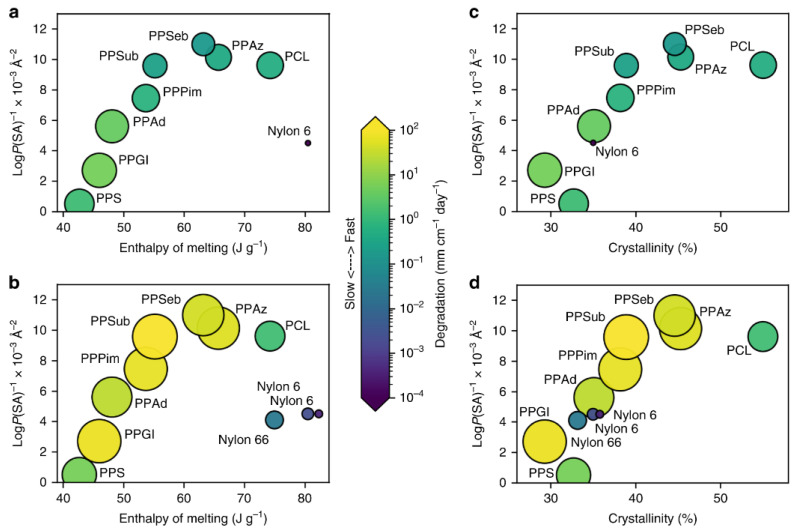
The influence of crystallinity and hydrophobicity on degradation [62]. The computational Log*P*(SA)^−1^ values are plotted versus the enthalpy of melting for abiotic hydrolysis (**a**), the enthalpy of melting for biotic processes (**b**), the % crystallinity for abiotic hydrolysis (**c**), the % crystallinity for biotic processes (**d**).The size of circles and colour is equivalent to surface erosion (in mg cm^−2^ day^−1^) in artificial seawater.

**Figure 14 polymers-14-00216-f014:**
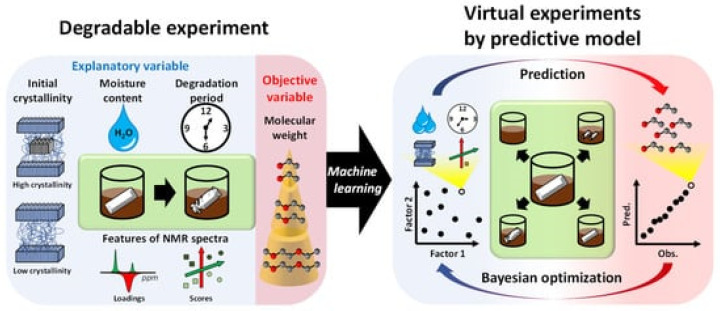
Real and virtual degradable experiments according to the study of Yamawaki et al. [265].

**Figure 15 polymers-14-00216-f015:**
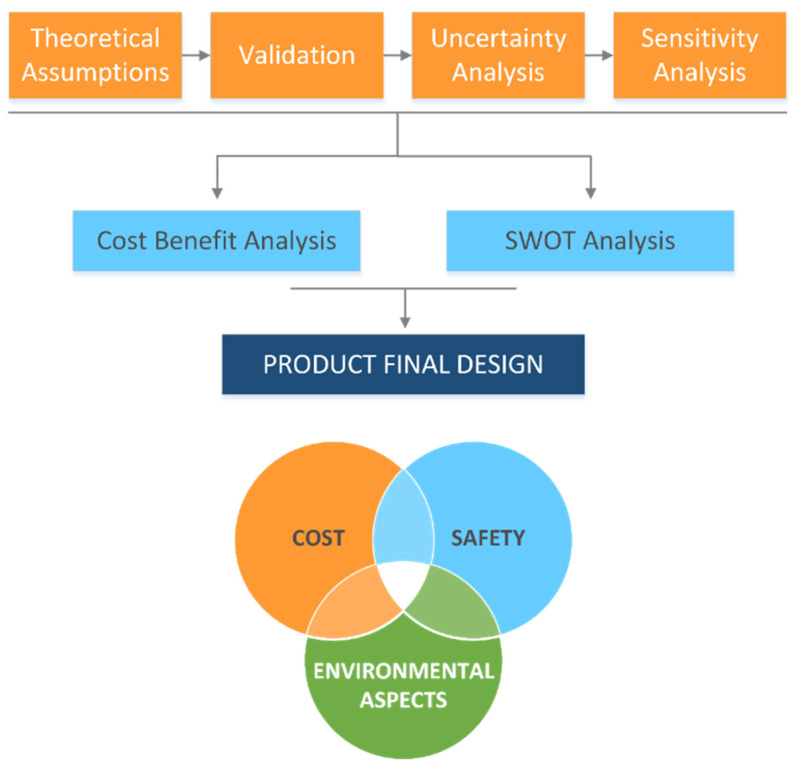
Modelling and Simulation Process for Products.

**Table 1 polymers-14-00216-t001:** Reinforcement comparison by fibre type, i.e., glass, carbon, basalt, aramid, and their market share, cost range and mechanical properties (in the unaged state), reproduced from [124].

Fibre Type ^1^	Market Share [%]	Cost Range [$/kg]	Tensile Strength [GPa]	Young’s Modulus [GPa]
E-Glass	~70%	1.3–2.6	3.45–3.5	72.5–73.5
E-CR-Glass		1.2–3	2–3.625	72.5–83
AR-Glass		2.5–3	1.7–3.5	72–175
C-Glass		1–2.5	3.3	69
A-Glass		2–3	3.3	72
S/S-2-Glass		16–26	4.6–4.9	86–89
R-Glass		16–26	4.4	86
PAN Type Carbon	~12%	15–120	1.8–7.0	230–540
HS Carbon		20–120	3.31–5	228–248
IM Carbon		25–120	4.1–6	265–320
HM Carbon		25–120	1.52–2.41	393–483
UHM Carbon		30–120	2.24	724
Basalt	~11%	5	4.84	89
Aramid/Kevlar	~7%	15–30	2.6–3.4	55–127

^1^ Fibre type abbreviations expanded: E-Glass [Electric], E-CR-Glass [Electric/Corrosion Resistant], AR-Glass [Alkali Resistant], C-Glass [Chemical], A-Glass [Alkali], S/S-2-Glass [Strength], R-Glass [Reinforcement], HS Carbon [High Strength], IM [Intermediate Modulus], HM Carbon [High Modulus], UHM Carbon [Ultra High Modulus].

**Table 2 polymers-14-00216-t002:** A condensed list of recent works modelling ageing in heterogeneous materials at multiple scales. Model abbreviations: DNS (Direct Numerical Simulation), AH (Analytical Homogenization), NH (Numerical Homogenization), CH (Computational Homogenization).

Ref(s)	Material(s)	Model	Process(es)
[194,195,196]	GFRP	DNS	Diffusion, swelling
[104]	CFRP	DNS	Diffusion, swelling
[197]	Flax/Epoxy	DNS	Diffusion, swelling
[198]	Woven composites	AH	Diffusion
[112]	GFRP	AH/NH	Swelling
[199]	GFRP	AH	Degradation
[200]	High-Vf polymers	AH/NH	Diffusion
[201]	GFRP	AH	Diffusion, swelling
[202]	GFRP	NH	Diffusion
[203]	GFRP	NH	Diffusion, degradation, fracture
[204]	Woven composites	NH	Diffusion
[205,206]	GFRP	NH	Diffusion, degradation
[207]	Concrete	CH	Diffusion, swelling, degradation, fracture
[208]	Polyamide composites	CH	Diffusion, swelling, degradation, damage
[202,209,210]	GFRP	CH	Diffusion, swelling, degradation, fracture
[211]	Titanium composites	Other	Diffusion, degradation
[212]	Braided composites	Other	Degradation, fracture

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
