# Peer review of "Modelling of Environmental Ageing of Polymers and Polymer Composites—Modular and Multiscale Methods"

_polymers, 2022, doi:10.3390/polym14010216_

Round 1

Reviewer 1 Report

Although I am a polymer scientist, I have almost no experience of modelling of these systems. With that in mind, I found the paper to be an interesting read and useful in that it provides the reader with entry points to the various modelling approaches applicable to the components on these composite materials. Having said that, my impression was that its scope was too broad and this tended to result in somewhat superficial overviews in each section. I would encourage the authors to consider a follow-up paper that focuses perhaps on the modelling of degradation processes in thermoplastic (and perhaps thermosetting) matrices i.e. the focus is on the polymers.

Reviewer 2 Report

This review mainly revisited applying the modular and multiscale modeling approaches to describe the environmental aging/degradation process of polymer and polymer composites materials. The whole paper is well written and very well logically organized. From the introduction of degradation concept and various degradation type for polymers with different application to the detailed numerical modeling of the environmental degradation process as well as the industrial outlook, this review provided a broad (chemical/physical) but relative complete coverage of the related information on this topic. I am sure the authors made great effort to polish the manuscript before submission. After reading through the manuscript, I didn’t find any issue in this paper. This is an interesting and informative review paper I suggest publishing as is.